# Osteoblast/osteocyte-derived interleukin-11 regulates osteogenesis and systemic adipogenesis

Bingzi Dong [1,2,3,10], Masahiro Hiasa[4,10], Yoshiki Higa [4], Yukiyo Ohnishi[2], Itsuro Endo [2], Takeshi Kondo[2], Yuichi Takashi [1], Maria Tsoumpra [1], Risa Kainuma[1,5], Shun Sawatsubashi [1], Hiroshi Kiyonari [6], Go Shioi[6], Hiroshi Sakaue [7], Tomoki Nakashima[8], Shigeaki Kato [9], Masahiro Abe[2], Seiji Fukumoto [1] & Toshio Matsumoto [1] ✉

Exercise results in mechanical loading of the bone and stimulates energy expenditure in the adipose tissue. It is therefore likely that the bone secretes factors to communicate with adipose tissue in response to mechanical loading. Interleukin (IL)−11 is known to be expressed in the bone, it is upregulated by mechanical loading, enhances osteogenesis and suppresses adipogenesis. Here, we show that systemic IL-11 deletion (IL-11$^{-/-}$) results in reduced bone mass, suppressed bone formation response to mechanical loading, enhanced expression of Wnt inhibitors, and suppressed Wnt signaling. At the same time, the enhancement of bone resorption by mechanical unloading was unaffected. Unexpectedly, IL-11$^{-/-}$ mice have increased systemic adiposity and glucose intolerance. Osteoblast/osteocyte-specific IL-11 deletion in osteocalcin-Cre;IL-11$^{fl/fl}$ mice have reduced serum IL-11 levels, blunted bone formation under mechanical loading, and increased systemic adiposity similar to IL-11$^{-/-}$ mice. Adipocyte-specific IL-11 deletion in adiponectin-Cre;IL-11$^{fl/fl}$ did not exhibit any abnormalities. We demonstrate that osteoblast/osteocyte-derived IL-11 controls both osteogenesis and systemic adiposity in response to mechanical loading, an important insight for our understanding of osteoporosis and metabolic syndromes.

Exercise offers mechanical loading to maintain bone remodeling balance and muscle strength, while it stimulates energy expenditure in the adipose tissue. As these responses are simultaneously and rapidly evoked, metabolic communications among tissues appear to underlie the systemic response. Because bone is an endocrine organ controlling mineral homeostasis[1,2] as well as energy expenditure[3], we assumed presence of an exercise sensor derived from bone to communicate with the adipose tissue for energy expenditure. We have previously found that interleukin (IL)-11 is expressed mainly in the bone and is upregulated by mechanical loading to bone[4]. Because the biological

[1]Fujii Memorial Institute of Medical Sciences, Tokushima University, Tokushima, Japan. [2]Department of Endocrinology, Metabolism and Hematology, Tokushima University Graduate School of Medical Sciences, Tokushima, Japan. [3]Department of Endocrinology and Metabolism, The Affiliated Hospital of Qingdao University, Qingdao, China. [4]Department of Orthodontics and Dentofacial Orthopedics, Tokushima University Graduate School of Dentistry, Tokushima, Japan. [5]Setsuro Tech Inc., Tokushima, Japan. [6]Laboratory for Animal Resources and Genetic Engineering, RIKEN Center for Biosystems Dynamics Research, Kobe, Japan. [7]Department of Nutrition and Metabolism, Tokushima University Graduate School of Nutritional Sciences, Tokushima, Japan. [8]Department of Cell Signaling, Tokyo Medical and Dental University, Tokyo, Japan. [9]Fukushima Medical University, Fukushima, Japan. [10]These authors contributed equally: Bingzi Dong, Masahiro Hiasa. ✉e-mail: toshio.matsumoto@tokushima-u.ac.jp

activity of IL-11 was originally reported to inhibit adipogenesis of bone marrow adipocytes in vitro, and designated also as adipogenesis inhibitory factor (AGIF)[5], there is a possibility that IL-11 may be the factor coupling exercise-induced changes in bone remodeling with those in adipose tissue metabolism.

IL-11 is one of interleukin members which were originally identified as cytokines secreted from helper T cells. Further studies have uncovered broader spectrum of biological actions of IL-11 including regulation of hematopoiesis[6], cancer development/progression[7], and mesenchymal cell differentiation[8]. As to its biological action on bone remodeling, a detailed dissection of IL-11 action showed that IL-11 enhanced osteoclast formation and bone resorption[9], and this activity was later shown to be mediated by stimulation of osteoblastic differentiation from mesenchymal progenitors[10]. In addition, recent reports

**Fig. 1 | Decreased bone mass with reduced bone formation without change in bone resorption by histomorphometric analysis of the vertebral bone in IL-11$^{-/-}$ mice. a** Micro-CT of femoral bones from 12-week-old WT and IL-11$^{-/-}$ mice. Scale bar indicates 1 mm. **b, c** Total, cortical and cancellous bone mineral density (BMD) of the vertebra from WT (open circle) ($n$ = 9, 10, 10, 11, 13) and IL-11$^{-/-}$ mice (closed circle) ($n$ = 14, 16, 16, 8, 7), and the femur from WT ($n$ = 18, 9, 9, 18, 12) and IL-11$^{-/-}$ mice ($n$ = 18, 15, 13, 16, 14) at 1, 2, 3, 4 and 6 months. Data are presented as means ± SD. *P* values are comparisons between WT and IL-11$^{-/-}$ mice using two-way ANOVA by Sidak's multiple comparisons test. **d** Total and cortical BMD of calvaria from 12-week-old WT ($n$ = 6, 6) and IL-11$^{-/-}$ mice ($n$ = 7, 5). Data are presented as means ± SE. **e** Bone histomorphometric analysis of vertebral bones from 12-week-old WT and IL-11$^{-/-}$

mice. BV/TV, bone volume/tissue volume of WT ($n$ = 6) and IL-11$^{-/-}$ mice ($n$ = 5); Tb.Th, trabecular thickness of WT ($n$ = 6) and IL-11$^{-/-}$ mice ($n$ = 5); Tb.N, trabecular number of WT ($n$ = 7) and IL-11$^{-/-}$ mice ($n$ = 5); Tb.Sp, trabecular separation of WT ($n$ = 7) and IL-11$^{-/-}$ mice ($n$ = 5); MAR, mineral apposition rate of WT ($n$ = 5) and IL-11$^{-/-}$ mice ($n$ = 5); BFR, bone formation rate of WT ($n$ = 5) and IL-11$^{-/-}$ mice ($n$ = 5); Oc.S/BS, osteoclast surface/bone surface of WT ($n$ = 7) and IL-11$^{-/-}$ mice ($n$ = 5); N.Oc/B.Pm, number of osteoclasts/bone perimeter of WT ($n$ = 6) and IL-11$^{-/-}$ mice ($n$ = 6); OS/BS, osteoid surface/bone surface of WT ($n$ = 7) and IL-11$^{-/-}$ mice ($n$ = 5); Ob.S/BS, osteoblast surface/bone surface of WT ($n$ = 7) and IL-11$^{-/-}$ mice ($n$ = 5); N.Ob/B.Pm, number of osteoblasts/bone perimeter of WT ($n$ = 7) and IL-11$^{-/-}$ mice ($n$ = 5). Data are means ± SE. *P* values are calculated by two-tailed Student's unpaired *t*-test.

demonstrate that IL-11 receptor α knockout (IL-11Rα$^{-/-}$) mice, but not IL-11 deficient mice, as well as several coding missense mutations in IL-11RA gene have been linked to craniosynostosis[11,12]. Thus, deficient IL-11 receptor signal appears to be associated with abnormalities in human bone development. Because IL-11Rα$^{-/-}$ mice exhibited increased bone mass with cell-autonomous reduction in osteoclast precursor differentiation[13], the physiological role of IL-11 for bone remodeling was considered as an osteoclastic regulator. However, on the contrary, when IL-11 was overexpressed in transgenic mice, bone formation and bone mass were unexpectedly increased without changes in osteoclastic bone resorption[14]. Moreover, these mice were resistant to age-related bone loss. Thus, controversy evidently remains as to the role of IL-11 on bone remodeling.

Mechanical loading to osteoblastic cells stimulates Ca$^{2+}$ influx which activates cAMP-responsive element binding protein (CREB) signaling via extracellularly regulated kinase (ERK) in osteoblasts, and activated CREB enhances ΔFosB expression[15], suggesting that ΔFosB is a sensing regulator of mechanical loading in gene regulatory cascade. As mechanical loading induces orchestral response in bone remodeling, a number of genes are conceivably responsive to ΔFosB expression. We have already reported that ΔFosB/JunD heterodimer binds to AP-1 sites on *IL-11* gene promoter and that indeed upregulates *IL-11* gene transcription with osteoblast differentiation in response to mechanical loading[4]. Gene expression analysis further found that Wnt signaling is enhanced via suppression of Wnt inhibitors by IL-11 stimulation, suggesting that the IL-11 response to mechanical loading in bone remodeling may result in activation of Wnt signaling[4]. The cascade of IL-11 stimulation to osteoblast differentiation is modulated by other factors. Parathyroid hormone acts as a positive hormone, while glucocorticoid and aging are negative factors[16,17]. Those past findings look favorable in the effect of IL-11 on bone formation, but the in vivo role of IL-11 in regulating osteoblast differentiation and bone formation remains elusive.

To illustrate the physiological impact of IL-11 in intact animals, a series of genetically manipulated mouse lines were generated. Systemic deletion of *IL-11* gene in mice caused a reduction in Wnt signaling and in bone mass with suppressed bone formation. To our surprise, *IL-11* gene deletion also caused an increase in the systemic white adipose tissue (WAT) with glucose intolerance. Therefore, we further created osteocalcin (Ocn)-Cre;IL-11$^{fl/fl}$ mice and adiponectin (Apn)-Cre;IL-11$^{fl/fl}$ mice to examine the role of osteoblast/osteocyte-derived and adipocyte-derived IL-11 actions on the regulation of bone and adipose tissue mass. Such tissue-specific manipulation of *IL-11* gene should define the role of osteoblast/osteocyte-derived IL-11 in stimulation of osteogenesis along with regulation of systemic adiposity and energy metabolism. Thus, the present study illustrates the physiologically significant role of osteoblast/osteocyte-derived IL-11 on not only skeletal but also adipose tissue mass.

## Results
### Reduced bone formation and bone mass in IL-11$^{-/-}$ mice
Total, cortical and cancellous bone mineral density (BMD) of the vertebra and femur of IL-11$^{-/-}$ mice was reduced after weaning (Fig. 1a–c).

BMD of calvaria, a non-weight-bearing bone, in IL-11$^{-/-}$ mice was almost the same as that in WT mice (Fig. 1d), similar to that in unloading conditions[18,19]. Vertebral histomorphometric analysis confirmed reduced bone mass in IL-11$^{-/-}$ mice with reduced bone formation and unchanged bone resorption (Fig. 1e).

### Reduced Wnt signaling in the bone of IL-11$^{-/-}$ mice
Quantitative PCR of femurs revealed that expression of osteogenic genes, *Ocn*, *Runx2* and *Osterix (Osx)* was reduced in IL-11$^{-/-}$ mice, while the expression of *receptor activator of NF-κB ligand (RANKL)* and *osteoprotegerin (OPG)* as well as *RANKL/OPG* ratio was unchanged and that of osteoclastic genes, *cathepsin K (Ctsk)* and *tartrate-resistant acid phosphatase (Trap)*, was not different from wild-type (WT) mice (Fig. 2a). Serum bone formation markers, osteocalcin (Ocn) and alkaline phosphatase (ALP), were reduced, whereas a bone resorption marker, TRAP, was not changed in IL-11$^{-/-}$ mice (Fig. 2b). Although the most closely related cytokine to IL-11 and has also an important role in bone metabolism is IL-6, serum IL-6 concentration in IL-11$^{-/-}$ mice was not different from that in WT mice (Fig. 2b). Expression of Wnt inhibitor genes, *Sost*, *Dickkopf1 (Dkk1)* and *Dkk2*, increased in the femur of IL-11$^{-/-}$ mice (Fig. 2c), and that of Wnt target genes, *Axin2*, *Cyclin D1 (Ccnd1)* and *Ccnd4*, was reduced (Fig. 2d). We backcrossed TOPGAL mice with IL-11$^{-/-}$ mice (TOPGAL;IL-11$^{-/-}$) to detect Wnt signaling in vivo. TOPGAL;IL-11$^{-/-}$ mice exhibited less X-gal positive osteoblasts and osteocytes compared to TOPGAL;WT mice (Fig. 2e). Immunostaining of Wnt inhibitors, sclerostin, Dkk1 and Dkk2 was increased in IL-11$^{-/-}$ mice (Fig. 2e). Thus, the enhanced expression of Wnt inhibitors in the bone was associated with reduced Wnt signaling, which appeared to play a role at least in part in the reduction of bone formation in IL-11$^{-/-}$ mice.

### IL-11Rα -STAT-HDAC signal suppresses Sost expression
In order to clarify whether the effects of IL-11 were mediated via its receptor, IL-11 receptor α (IL-11Rα), the signaling pathway of IL-11 in osteoblastic cells was examined using control and IL-11Rα knockout (KO) MC3T3-E1 cells (Supplementary Fig 1a–c). In control MC3T3-E1 cells, *Sost* mRNA expression increased with differentiation of MC3T3-E1 cells cultured in osteogenic medium (Supplementary Fig 2). IL-11 treatment enhanced phosphorylation of STAT1 and STAT3 within 15 min. In contrast, phosphorylation of both STAT1 and 3 was abolished in IL-11Rα$^{-/-}$ cells (Fig. 2f). In IL-11Rα$^{-/-}$ cells, the expression of *Rankl* and *Opg* mRNA was not different from that in control MC3T3-E1 cells. However, *Sost* mRNA was higher in IL-11Rα$^{-/-}$ cells compared to that in control cells (Fig. 2g). Because STAT3 is reported to play an important role in regulating osteogenesis[20], and because class IIa histone deacetylases, HDAC4 and HDAC5, are shown to be required for loading-induced Sost suppression and bone formation[21], we next examined whether nuclear translocation of HDAC4/5 is enhanced by IL-11 and whether STAT3 inhibition influences nuclear translocation of HDAC4/5 and Sost expression in MC3T3-E1 cells. As shown in Fig. 2h, i, IL-11 enhanced nuclear translocation of HDAC4/5, and a STAT3 inhibitor, STAT3-IN-1, abrogated IL-11-induced nuclear translocation of

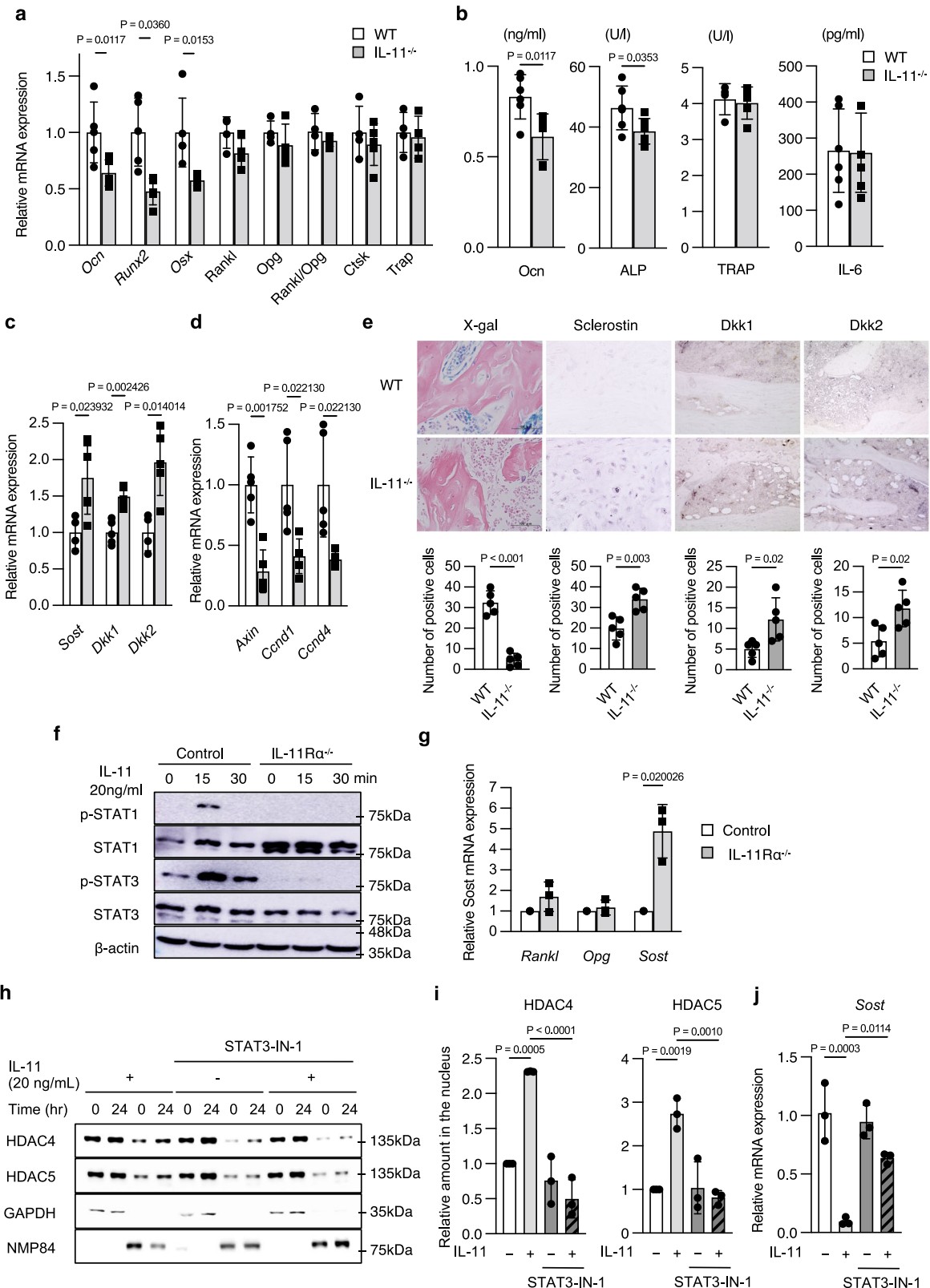

HDAC4/5. In parallel with the inhibition of HDAC4/5 translocation, inhibition of STAT3 by STAT3-IN-1 abrogated the suppression of *Sost* expression by IL-11 (Fig. 2j). These results demonstrate that IL-11 acts on osteoblastic cells via IL-11Rα to enhance STAT1/3 phosphorylation, which increases nuclear translocation of HDAC4/5 and suppresses *Sost* gene expression without affecting *Rankl* or *Opg* expression.

## Mechanical loading enhances IL-11 expression in the bone

Because mechanical loading enhanced IL-11 expression in osteoblasts[4], the role of IL-11 on unloading and reloading-induced changes in bone was examined using IL-11$^{-/-}$ mice. After tail suspension, vertebral cancellous BMD decreased in both WT and IL-11$^{-/-}$ mice (Fig. 3a). After reloading, vertebral cancellous BMD of WT mice increased to ground control level, whereas that of IL-11$^{-/-}$ mice did not reach ground control

**Fig. 2 | Decreased expression of osteoblastic genes with enhanced expression of Wnt inhibitors and reduced Wnt signaling in the femoral bone, and reduced serum bone formation markers in IL-11$^{-/-}$ mice. a** Expression of osteoblastic and osteoclastic genes in femur of 12-week-old WT and IL-11$^{-/-}$ mice. *Ocn*, osteocalcin; *Runx2*; *Osx*, osterix in WT ($n = 5$) and IL-11$^{-/-}$ mice ($n = 5$); *Rankl*, receptor activator of *NF-kB ligand; Opg*, osteoprotegerin; *Trap*, tartrate-resistant acid phosphatase in WT ($n = 4$) and IL-11$^{-/-}$ mice ($n = 4$); *Ctsk*, *Cathepsin K* in WT ($n = 4$) and IL-11$^{-/-}$ mice ($n = 5$). Data are mean ± SD. *P* values are calculated by two-tailed Student's unpaired *t*-test. **b** Serum bone turnover markers. OCN, osteocalcin in WT ($n = 6$) and IL-11$^{-/-}$ mice ($n = 6$); ALP, alkaline phosphatase in WT ($n = 6$) and IL-11$^{-/-}$ mice ($n = 7$); TRAP, tartrate-resistant acid phosphatase in WT ($n = 4$) and IL-11$^{-/-}$ mice ($n = 5$). Serum IL-6 in WT ($n = 6$) and IL-11$^{-/-}$ mice ($n = 6$). Data are means ± SD. *P* values are calculated by two-tailed Student's unpaired *t*-test. **c** Messenger RNA expression of Wnt inhibitors. *Sost* in WT ($n = 4$) and IL-11$^{-/-}$ mice ($n = 6$); *Dkk1, Dickkopf1* in WT ($n = 5$) and IL-11$^{-/-}$ mice ($n = 5$); *Dkk2* in WT ($n = 4$) and IL-11$^{-/-}$ mice ($n = 5$). Data are means ± SD. P values are calculated by two-tailed Student's unpaired *t*-tests. **d** Wnt target genes in femoral bones from 12-week-old WT ($n = 5$) and IL-11$^{-/-}$ mice ($n = 5$). *Ccnd1, Cyclin D1.* Data are means ± SD. P values are calculated by two-tailed Student's unpaired *t*-tests. **e** X-gal staining and immunostaining for sclerostin, Dkk1 and 2 of tibia from

10-week-old WT and IL-11$^{-/-}$ mice. Scale bar indicates 100 µm. Figures in the bottom panel demonstrate numbers of stained cells in each microscopic field at ×100 magnification. Data are means of triplicate counts in each mouse from 5 mice and expressed as means ± SD. *P* values are calculated by two-tailed Student's unpaired t-test. **f** STAT1,3 phosphorylation after stimulation with 20 ng/ml IL-11 for 0, 15 and 30 min. A representative figure of three independent experiments. **g** Expression of *Rankl, Osteoprotegerin (Opg)* and *Sost* genes in IL-11Rα knockout ($n = 3$) and control MC3T3-E1 cells ($n = 3$). Data are means ± SD. *P* values are calculated by two-tailed Student's unpaired *t*-tests. **h** Effect of 20 ng/mL IL-11 on nuclear translocation of HDAC4 and 5 in the presence or absence of 10 µM STAT-IN-1, a STAT3 inhibitor. A representative figure of three independent experiments. **i** Densitometric measurement of HDAC4 and 5 accumulated in the nucleus of MC3T3-E1 cells 24 h after 20 ng/mL IL-11 treatment in the presence ($n = 3$) or absence ($n = 3$) of 10 µM STAT-IN-1. The results are expressed as a relative amount of HDAC4/5 accumulated in the nucleus to that without IL-11 treatment. Data are means ± SD. *P* values are calculated by ordinary one-way ANOVA with Sidak's multiple comparisons test. **j** Effect of 20 ng/mL IL-11 on *Sost* mRNA expression in the presence ($n = 3$) or absence ($n = 3$) of 10 µM STAT-IN-1. Data are means ± SD. *P* values are calculated by ordinary one-way ANOVA with Sidak's multiple comparisons test.

level (Fig. 3a, b). Expression of *Sost*, *Dkk1* and *Dkk2* genes increased by unloading and returned to the baseline by reloading in WT mice, while expression of these genes was higher at baseline and did not change by unloading or reloading in IL-11$^{-/-}$ mice (Fig. 3c). Bone histomorphometric analysis revealed that bone formation parameters were suppressed and bone resorption parameters were elevated by unloading in both WT and IL-11$^{-/-}$ mice. After reloading, although bone resorption parameters decreased and bone formation parameters increased to baseline level in WT mice, bone formation parameters did not recover fully while bone resorption parameters decreased to baseline level in IL-11$^{-/-}$ mice (Fig. 3d). Serum IL-11 in WT mice was reduced after unloading, and recovered to baseline level after reloading (Fig. 3e). Because *IL-11* expression was much higher in bone than in other tissues (Supplementary Fig. 3), serum IL-11 appeared to originate mostly from bone. These data demonstrate that mechanical loading enhances the production of IL-11 in the bone which suppresses the expression of Wnt inhibitors, while in IL-11$^{-/-}$ mice the production of IL-11 in the bone is not increased by mechanical loading and bone formation response to mechanical reloading is blunted.

**Increased systemic adiposity in IL-11$^{-/-}$ mice**

In IL-11$^{-/-}$ mice, bone marrow adiposity increased in a time-dependent manner (Fig. 4a), and adipocytic differentiation of bone marrow stromal cells (BMSC) was enhanced (Fig. 4b, c). Addition of IL-11 suppressed adipocytic differentiation of BMSC in WT and IL-11$^{-/-}$ mice (Fig. 4b, c). Osteoblast differentiation of BMSC from IL-11$^{-/-}$ mice was reduced, and expression of osteogenic genes, *Ocn, Runx2* and *Osx*, decreased, while that of adipogenic genes, *peroxisome proliferator-activated receptor γ (Pparγ)* and *CCAAT enhancer binding protein α (Cebpα)* increased in BMSC from IL-11$^{-/-}$ mice (Fig. 4d). In addition to the increase in bone marrow adiposity in IL-11$^{-/-}$ mice, adipose tissue (AT) weight was higher in IL-11$^{-/-}$ mice under both regular diet (RD) and high-fat diet (HFD) (Supplementary Fig. 4), and total, visceral and subcutaneous AT area was larger (Figs. 4e, f, Supplementary Fig. 5). Body weight gain and size were not significantly different between WT and IL-11$^{-/-}$ mice (Supplementary Fig. 6a, b), and weight of other tissues was not different between WT and IL-11$^{-/-}$ mice (Supplementary Fig. 4). Serum leptin and adiponectin levels were similar under RD in WT and IL-11$^{-/-}$ mice, but IL-11$^{-/-}$ mice showed higher leptin and lower adiponectin levels than WT mice under HFD (Fig. 4g). Adipocyte diameter and relative number increased in IL-11$^{-/-}$ mice (Fig. 4h, Supplementary Fig. 7a, b).

**Suppressed Wnt signal in the adipose tissue of IL-11$^{-/-}$ mice**

Brown AT (BAT) weight and expression of BAT-specific genes, *Pparγ coactivator (Pgc)-1α* and *uncoupling protein (Ucp)-1*, were not different

between WT and IL-11$^{-/-}$ mice (Fig. 5a). Expression of lipolytic enzyme genes, *hormone-sensitive lipase (Hsl)* and *adipose triglyceride lipase* (*Atgl*), in white adipose tissue (WAT) decreased in IL-11$^{-/-}$ mice, while expression of fatty acid β-oxidation enzyme, *acyl-CoA oxidase (Aco)*, and lipogenic enzyme, *acetyl-CoA carboxylase (Acc)*, was similar in both mice (Fig. 5b).

X-gal staining of tissues from TOPGAL;WT and TOPGAL;IL-11$^{-/-}$ mice indicated reduced Wnt signaling in the AT in IL-11$^{-/-}$ mice (Fig. 5c). Expression of *Dkk1* and *Dkk2* mRNA in WAT increased in IL-11$^{-/-}$ mice, and decreased by HFD in WT but not in IL-11$^{-/-}$ mice (Fig. 5d), demonstrating that Wnt signaling was suppressed in WAT of IL-11$^{-/-}$ mice via enhanced Wnt inhibitor expression. Among inflammatory cytokines, *monocyte chemoattractant protein 1 (MCP1)* mRNA expression in WAT was higher in IL-11$^{-/-}$ mice, and HFD further enhanced *MCP1* expression in IL-11$^{-/-}$ mice. *Tumor necrosis factor α (TNFα)* mRNA expression in WAT was higher in IL-11$^{-/-}$ than in WT mice under HFD, and HFD enhanced *TNFα* expression in both WT and IL-11$^{-/-}$ mice (Fig. 5e). Blood glucose was higher by oral glucose tolerance test (oGTT), and area under the curve (AUC) was larger in both RD and HFD-fed IL-11$^{-/-}$ mice (Fig. 5f). Fasting level and AUC of serum insulin in oGTT as well as homeostasis model assessment of insulin resistance (HOMA-IR) were higher in IL-11$^{-/-}$ mice (Fig. 5g). These results altogether indicate that IL-11$^{-/-}$ mice develop insulin resistance and glucose intolerance with increased WAT.

**IL-11 deletion in osteoblasts recapitulates IL-11$^{-/-}$ mice**

To clarify whether reduced bone formation and increased adiposity were due to reduced IL-11 expression in the bone or in the adipose tissue, we created conditional IL-11 knockout mice by crossing *osteocalcin* gene promoter-driven (Ocn-Cre) or *adiponectin* gene promoter-driven Cre recombinase transgenic mice (Apn-Cre) with *IL-11* gene floxed (IL-11$^{fl/fl}$) mice (Supplementary Fig. 8, Fig. 6a). Osteoblast/osteocyte-specific *IL-11* gene deletion in Ocn-Cre;IL-11$^{fl/fl}$ mice showed reduced bone mass with decreased cortical and cancellous BMD similar to conventional IL-11$^{-/-}$ mice (Fig. 6b, c), whereas adipocyte-specific *IL-11* gene deletion in Apn-Cre;IL-11$^{fl/fl}$ mice showed no difference in cortical or cancellous BMD from those in control IL-11$^{fl/fl}$ mice (Fig. 6b, c). Serum IL-11 and osteocalcin levels were reduced in Ocn-Cre;IL-11$^{fl/fl}$ mice but did not change in Apn-Cre;IL-11$^{fl/fl}$ mice (Fig. 6d). Serum TRAP level was similar in all the groups (Fig. 6d). Expression of osteogenic genes, *Ocn, Runx2* and *Osterix* was suppressed, but of *Rankl, Opg* and *Rankl/Opg* ratio as well as osteoclastic genes, *Ctsk* and *TRAP*, was unchanged in Ocn-Cre;IL-11$^{fl/fl}$ mice (Fig. 6e). Expression of those genes was not different from that of control IL-11$^{fl/fl}$ mice in Apn-Cre;IL-11$^{fl/fl}$ mice (Fig. 6e). Furthermore, while mechanical unloading reduced vertebral

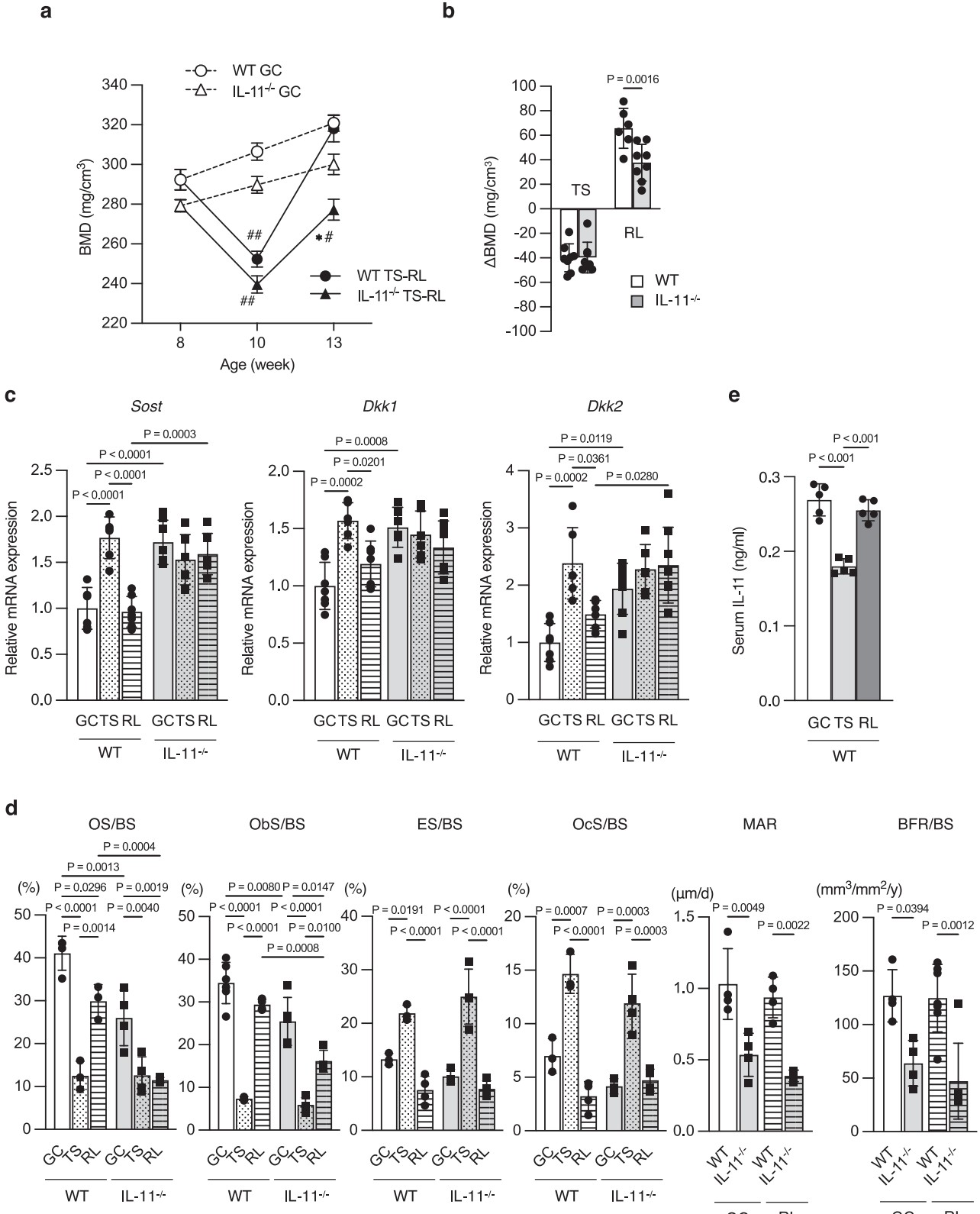

cancellous BMD in both control IL-11^fl/fl and Ocn-Cre;IL-11^fl/fl mice, mechanical reloading increased BMD to a lesser extent in Ocn-Cre;IL-11^fl/fl mice compared to that in control IL-11^fl/fl mice (Fig. 6f, Supplementary Fig. 9). Thus, osteoblast specific IL-11 deletion in Ocn-Cre;IL-11^fl/fl mice recapitulated all the bone phenotypes of conventional IL-11^−/− mice. Furthermore, AT area under HFD was larger in Ocn-Cre;IL-11^fl/fl than in control IL-11^fl/fl mice, but was unchanged in

Apn-Cre;IL-11^fl/fl mice (Fig. 6g, h). Thus, the increase in AT mass by systemic deletion of IL-11 was due to a reduction in osteoblast/osteocyte-derived IL-11 but not adipocyte-derived IL-11. Blood glucose was higher after glucose loading by oGTT, and AUC was larger than control IL-11^fl/fl mice in Ocn-Cre;IL-11^fl/fl but not in Apn-Cre;IL-11^fl/fl mice under both RD and HFD (Fig. 6i). Fasting serum insulin as well as HOMA-IR were higher in Ocn-Cre;IL-11^fl/fl mice compared with control IL-11^fl/fl and

**Fig. 3 | Normal enhancement of bone resorption in response to mechanical unloading, and reduced bone formation in response to mechanical reloading with sustained elevation of Wnt inhibitors in IL-11$^{-/-}$ mice. a** Vertebral BMD in ground control (GC) group of WT ($n = 7$) and IL-11$^{-/-}$ mice ($n = 7$), tail suspension (TS) group of WT ($n = 8$) and IL-11$^{-/-}$ mice ($n = 8$) and reloading (RL) group of WT ($n = 6$) and IL-11$^{-/-}$ mice ($n = 8$). Data are means ± SE. *$P < 0.0001$ vs WT, and $^{\#}P = 0.0217$, $^{\#\#}P < 0.0001$ vs GC group in the same genotype using one-way ANOVA with Tukey's multiple comparisons test. **b** BMD change after TS in WT (open bar) ($n = 8$) and IL-11$^{-/-}$ mice (closed bar) ($n = 8$), and after RL in WT ($n = 6$) and IL-11$^{-/-}$ mice ($n = 8$). Data are means ± SD. $P$ values are calculated by two-way ANOVA with Bonferroni's multiple comparisons test. **c** Expression of Wnt inhibitors in femoral bones from WT in GC ($n = 6$), TS ($n = 6$) and RL ($n = 7$) goups, and from IL-11$^{-/-}$ mice in GC ($n = 7$), TS ($n = 6$) and RL ($n = 6$) goups. Data are means ± SD. $P$ values are calculated using two-way ANOVA with Tukey's multiple comparisons test. **d** Bone histomorphometric analysis of the vertebrae of WT and IL-11$^{-/-}$ mice. OS/BS, osteoid surface/bone surface of WT in GC, TS and RL group ($n = 3$), and of IL-11$^{-/-}$ mice in GC, TS and RL group ($n = 4$); Ob.S/BS, osteoblast surface/bone surface of WT in GC ($n = 6$), TS ($n = 3$) and RL group ($n = 4$), and of IL-11$^{-/-}$ mice in GC ($n = 5$), TS ($n = 4$) and RL group ($n = 4$); ES/BS, eroded surface/bone surface of WT in GC ($n = 3$), TS ($n = 3$) and RL group ($n = 4$), and of IL-11$^{-/-}$ mice in GC ($n = 3$), TS ($n = 4$) and RL group ($n = 4$); Oc.S/BS, osteoclast surface/bone surface of WT in GC ($n = 3$), TS ($n = 3$) and RL group ($n = 4$), and of IL-11$^{-/-}$ mice in GC ($n = 3$), TS ($n = 4$) and RL group ($n = 4$); MAR, mineral apposition rate of WT and IL-11$^{-/-}$ mice ($n = 4$); BFR/BS, bone formation rate of WT in GC ($n = 4$) and RL ($n = 7$), and of IL-11$^{-/-}$ mice in GC ($n = 4$) and RL ($n = 6$). Data are means ± SD. $P$ values are calculated using two-way ANOVA with Tukey's multiple comparisons test. **e** Serum IL-11 level of WT mice in GC, TS and RL groups. $n = 5$. Data are means ± SD. $P$ values are calculated using two-way ANOVA with Tukey's multiple comparisons test.

Apn-Cre;IL-11$^{fl/fl}$ mice under RD and HFD (Fig. 6j). These results altogether indicate that only Ocn-Cre;IL-11$^{fl/fl}$ but not Apn-Cre;IL-11$^{fl/fl}$ mice develop insulin resistance and glucose intolerance with increased WAT.

## IL-11 deletion in osteoblasts increases Dkk in adipocytes

In order to clarify whether the increased adiposity of Ocn-Cre;IL-11$^{fl/fl}$ mice is mediated by a suppression of Wnt signaling in the adipose tissue, we next examined the effect of a β-catenin agonist[22] and a Dkk1 inhibitor[23], WAY-262611, on the adiposity of control IL-11$^{fl/fl}$ and Ocn-Cre;IL-11$^{fl/fl}$ mice fed either RD or HFD. Adipose tissue area was larger in mice under HFD than under RD, and was significantly larger in Ocn-Cre;IL-11$^{fl/fl}$ mice than in control mice both under RD and HFD. Treatment with WAY-262611 reduced the adipose tissue area in both control IL-11$^{fl/fl}$ and Ocn-Cre;IL-11$^{fl/fl}$ mice under HFD (Figs. 7a, b). In mice under RD, because the adipose tissue area in vehicle-treated control mice was small, WAY-262611 significantly reduced adipose tissue area only in Ocn-Cre;IL-11$^{fl/fl}$ mice (Fig. 7b). In the adipose tissue, markedly enhanced expression of *CEBPα* in both control and Ocn-Cre;IL-11$^{fl/fl}$ mice under HFD was suppressed by WAY-262611 treatment (Fig. 7c). *Dkk2* expression in the adipose tissue was enhanced in Ocn-Cre;IL-11$^{fl/fl}$ mice under both RD and HFD, and WAY-262611 treatment did not affect the expression of *Dkk2*. The expression of a Wnt target gene, *Axin*, was markedly enhanced by WAY-262611 treatment in both control and Ocn-Cre;IL-11$^{fl/fl}$ mice regardless of whether they were under RD or HFD (Fig. 7c). *Sost* expression in the bone was enhanced in Ocn-Cre;IL-11$^{fl/fl}$ mice, but was not expressed in the adipose tissue (Fig. 7d).

## IL-11 enhances Dkk expression via IL-11Rα in adipocytes

In order to find out whether the increase in the adiposity of Ocn-Cre;IL-11$^{fl/fl}$ mice was mediated by a direct effect of IL-11 on adipocytes via IL-11Rα, we created an IL-11Rα$^{-/-}$ C3H10T1/2 cell line. When IL-11 was added to WT C3H10T1/2 cells, a dose-dependent suppression of adipogenesis was observed when cells were cultured in an adipogenic medium. In contrast, adipogenic differentiation was enhanced regardless of the presence or absence of IL-11 in IL-11Rα$^{-/-}$ cells (Fig. 7e). In parallel with those observations, mRNA expression of *PPARγ* and *CEBPα* was suppressed by IL-11 in WT C3H10T1/2 cells, but was markedly enhanced in IL-11Rα$^{-/-}$ cells, and the addition of IL-11 was without effect (Fig. 7f). The addition of IL-11 suppressed the expression of *Dkk1* and *2* in WT cells, whereas the expression of those Wnt inhibitors was markedly enhanced in IL-11Rα$^{-/-}$ cells, and IL-11 was unable to suppress the expression of *Dkk1* and *2* (Fig. 7f). Because *Sost* is not expressed in the adipocytes (Fig. 7d), these results indicate that IL-11 directly acts on the adipocytes via IL-11Rα to suppress adipogenesis via enhancement of Wnt signaling by suppressing *Dkk1* and *2*.

## Discussion

The present results unexpectedly demonstrate that adipogenic differentiation is enhanced not only in the bone marrow but also in the extra-skeletal adipose tissue in systemic IL-11$^{-/-}$ and Ocn-Cre;IL-11$^{fl/fl}$ mice. To our knowledge, no previous reports with loss of IL-11 signaling such as IL-11Rα$^{-/-}$ or gp130$^{-/-}$ mice demonstrated changes in the adipose tissue[13,24]. Because canonical Wnt signal suppresses adipogenesis and enhances osteoblastogenesis[8], and because IL-11 deletion in bone enhances the expression of Wnt inhibitors and suppresses Wnt signaling not only in the bone but also in the adipose tissue, these results suggest that IL-11 acts as an upstream modulator in the expression of Wnt inhibitors to cause these changes. However, the present results do not rule out the possibility that other mediators downstream IL-11 also play a role in regulating osteoblastogenesis and bone formation.

In regard to the role of Wnt signal in the adipose tissue, it was reported that Sost$^{-/-}$ mice exhibited a reduction in adipose tissue accumulation with increased insulin sensitivity, along with a dramatic increase in bone volume[25]. In contrast, sclerostin overproduction by adeno-associated virus transfection in the liver of mice resulted in the opposite phenotype with adipocyte hypertrophy. Those results suggested an endocrine function of sclerostin to regulate adipose tissue metabolism[25]. In addition, targeted deletion of TCF7L2, a TCF family key intracellular effector of the Wnt signaling, in adipocytes was shown to promote adipocyte hypertrophy and impaired glucose metabolism[26]. In humans, the administration of anti-sclerostin antibody, romosozumab, to patients with osteoporosis markedly increased bone formation and bone mass but there has been no report that demonstrated an effect of romosozumab in improving obesity. The present study demonstrated that the expression of other Wnt inhibitors, Dkk1, 2, in the adipose tissue was also upregulated in IL-11$^{-/-}$ mice. Taken together, those previous observations along with the present results are consistent with the notion that Wnt signal downstream IL-11 plays an important role in the negative regulation of adipose tissue mass. Although the present observations do not rule out the possibility that IL-11 may regulate other factor(s) which may also affect adipose tissue and bone metabolism, elucidation of the downstream mediators of IL-11 is out of the scope of this study. Further studies are needed to clarify this issue.

In the absence of IL-11, WAT is increased with increased size and number of adipocytes with reduced expression of lipolytic enzymes. As in human obesity, the resultant increase in adipose tissue mass enhances the expression of inflammatory cytokines in the adipose tissue, and appears to be responsible for the development of insulin resistance and glucose intolerance. Thus, systemic deletion of IL-11 causes not only a reduction in bone formation with increased adiposity in the bone marrow, but also an increase in systemic adiposity leading to obesity and insulin resistance. These features are reminiscent of those in a family with a missense mutation in LRP6, which developed metabolic syndrome and osteoporosis due to impaired canonical Wnt signaling[27].

In the present study, serum IL-11 is reduced and adipose tissue mass is increased not only in systemic IL-11$^{-/-}$ mice but also in Ocn-Cre;IL-11$^{fl/fl}$ mice, indicating that osteoblast/osteocyte-derived IL-11 acts

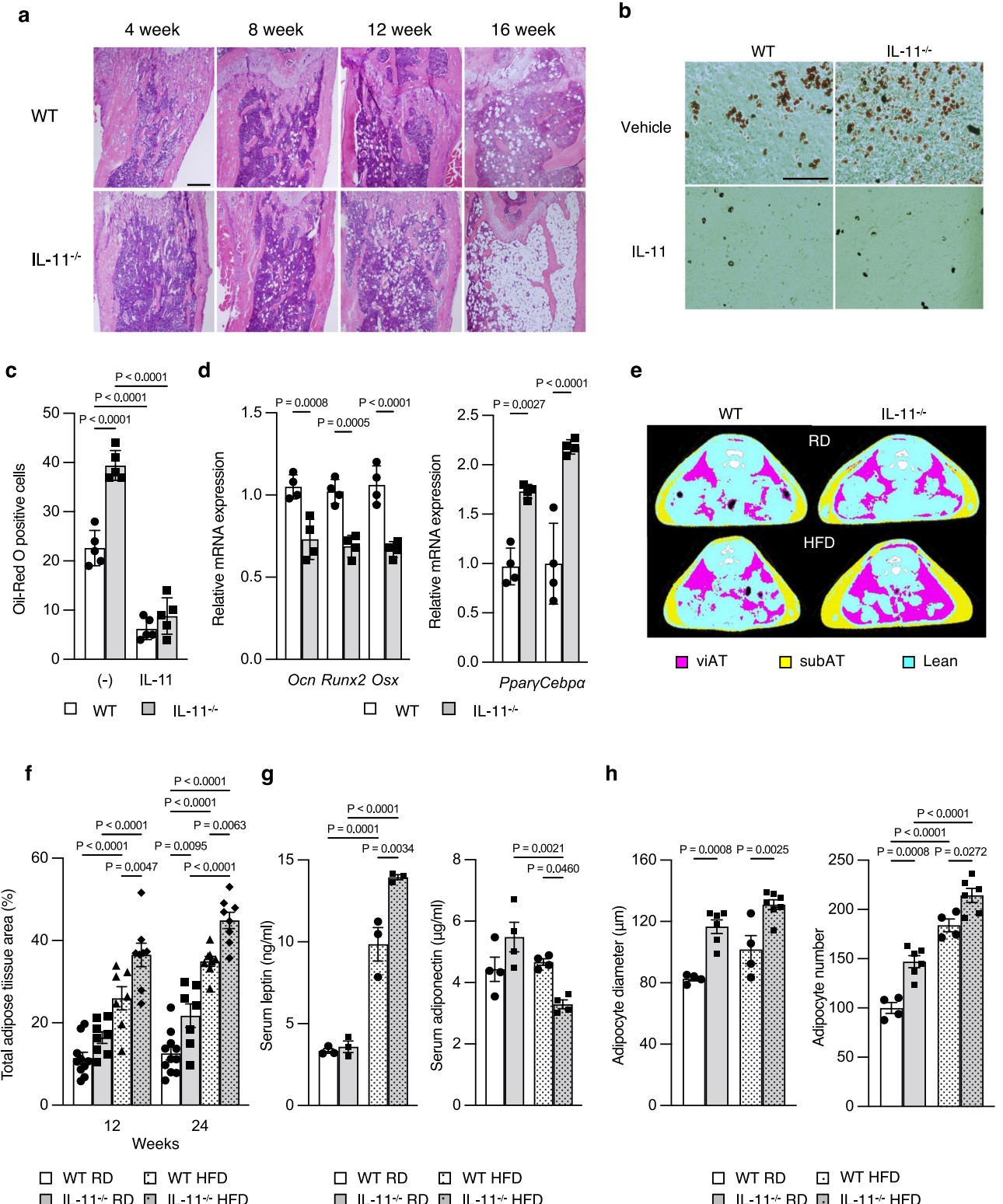

systemically to regulate adipose tissue mass. These results are another example of systemic hormonal action of bone-derived cytokines such as fibroblast growth factor 23 (FGF23)[1] and osteocalcin[3]. Thus, exercise stimulates IL-11 expression in the bone via enhanced mechanical loading, which enhances bone formation via, at least in part, the suppression of Wnt inhibitors. At the same time, because serum IL-11 is reduced by unloading and increased by reloading, increased bone-derived IL-11 in response to exercise acts like a hormone via systemic

circulation to control adipose tissue mass. Thus, serum IL-11 may be used to evaluate whether or not a subject takes enough exercise.

The present observations demonstrate that, in systemic IL-11$^{-/-}$ mice, BMD of weight-bearing bones was reduced by a reduction in bone formation without change in bone resorption. These results establish that the primary action of IL-11 in osteoblastic cells is to enhance bone formation without affecting osteoclastic bone resorption. A previous report demonstrated that systemic IL-11Rα$^{-/-}$

**Fig. 4 | Enhanced systemic adipogenesis along with enhanced adipogenic differentiation with reduced osteogenic differentiation of BMSCs in IL-11⁻/⁻ mice. a** H.E. staining of tibia from 4, 8, 12, and 16-week-old WT and IL-11⁻/⁻ mice. Scale bar = 500 μm. Representative pictures of three independent experiments. **b, c** Oil-Red O staining of bone marrow stromal cells (BMSC) from 12-week-old WT and IL-11⁻/⁻ mice. Scale bar = 100 μm. Adipogenesis was induced by $10^{-6}$ M troglitazone, with vehicle or 50 ng/ml recombinant IL-11. $n = 5$. Data are means ± SD. P value is calculated using two-way ANOVA with Tukey's multiple comparisons test. **d** Expression of osteogenic and adipogenic genes in BMSC from WT and IL-11⁻/⁻ mice. *Ppary, peroxisome proliferator-activated receptor γ; Cebpα, CCAAT enhancer binding protein α*. $n = 4$. Data are means ± SD. P values are calculated using two-way ANOVA with Tukey's multiple comparisons test. **e** Micro-CT analysis at L5 level of WT and IL-11⁻/⁻ mice on regular (RD) or high-fat diet (HFD) at 24 weeks. Pink, visceral AT (viAT);

yellow, subcutaneous AT (subAT); blue, lean mass; white, vertebral bone. **f** Quantitative analysis of total adipose tissue area at 12 weeks of WT on RD ($n = 10$) or HFD ($n = 7$) and of IL-11⁻/⁻ mice on RD ($n = 8$) or HFD ($n = 8$). Total adipose tissue area at 24 weeks of WT on RD ($n = 11$) or HFD ($n = 8$) and of IL-11⁻/⁻ mice on RD ($n = 7$) or HFD ($n = 8$). Data are means ± SE. P values are calculated using two-way ANOVA with Tukey's multiple comparisons test. **g** Fasting serum leptin ($n = 3$) and adiponectin ($n = 4$) in WT and IL-11⁻/⁻ mice on RD or HFD. P values are calculated using two-way ANOVA with Tukey's multiple comparisons test. **h** Adipocyte diameter in WAT from 32-week-old WT ($n = 4$) and IL-11⁻/⁻ mice on RD ($n = 6$) or HFD ($n = 7$). Relative adipocyte number in WAT from 32-week-old WT ($n = 4$) and IL-11⁻/⁻ mice ($n = 6$). Data are means ± SD. P values are calculated using two-way ANOVA with Tukey's multiple comparisons test.

---

mice exhibited reduced bone remodeling due to suppressed osteoclast differentiation[13]. In those mice, although bone remodeling was low and all the histomorphometric indices of bone formation were reduced, the alteration was regarded as not a cell-autonomous change in osteoblasts but was dependent on communications from other cells[13].

A later report demonstrated, using osteocyte-specific gp130 deletion in Dmp1-Cre;gp130fl/fl mice, that osteocytic gp130 signaling is required for maintaining bone formation and trabecular bone mass[24]. Actions of all IL-6-type cytokines, including IL-11, are mediated through their common receptor subunit gp130 to potently activate STAT3, and to a lesser extent STAT1[28]. In the present study, wild-type MC3T3-E1 osteoblastic cells respond to IL-11 by enhanced phosphorylation of STAT3 and to a lesser extent STAT1, while IL-11Rα-deficient MC3T3-E1 cells do not respond to IL-11 in STAT1/3 phosphorylation, and show enhanced expression of *Sost* without change in *Opg* or *Rankl* expression. These results demonstrate that IL-11 signal in osteoblasts is mediated at least in part via IL-11Rα-gp130-STAT1/3 signaling. The present results demonstrated that IL-11 enhanced nuclear translocation of class IIa HDAC, HDAC4/5, and a STAT3 inhibitor, STAT3-IN-1, inhibited the nuclear translocation of HDAC4/5 and abrogated the suppression of *Sost* expression by IL-11. These results suggest that IL-11 acts via IL-11Rα-gp130-STAT1/3 signaling to enhance translocation of HDAC4/5 and suppresses the expression of *Sost* in the bone.

We previously demonstrated that mechanical unloading suppressed and reloading enhanced *IL-11* gene expression in the hindlimb of mice in vivo. The signaling cascade to enhance *IL-11* expression by mechanical loading was examined in vitro. Fluid shear stress to mouse primary osteoblasts enhanced $Ca^{2+}$ influx via gadolinium-sensitive cation channel, which activated CREB via phosphorylation of ERK1/2. Activated CREB enhanced *fosB* gene transcription and increased ΔFosB expression[15]. A previous report demonstrated that ΔFosB transgenic mice exhibited enhanced bone formation and increased bone mass[29]. The increased ΔFosB formed complex with JunD on the AP-1 site of *IL-11* gene promoter, and enhanced *IL-11* transcription[4]. Mechanical loading increased IL-11 and suppressed the expression of Wnt inhibitors in osteoblastic cells, while mechanical unloading reduced IL-11 expression and enhanced the expression of Wnt inhibitors[4]. In the present study, the expression of Wnt inhibitors is not suppressed by mechanical loading in IL-11⁻/⁻ mice, and bone formation in response to mechanical loading is blunted in association with sustained high expression of Wnt inhibitors. These results indicate that IL-11 is a mechano-sensitive cytokine in the bone and plays an important role in regulating bone formation in response to mechanical loading. However, it is still unclear if the stimulation of IL-11 expression in bone is mediated via other mechano-sensors, such as cilia, integrins, G-proteins, Piezo-1 and other calcium channels. In addition, the present study does not rule out the possibility that the effect of IL-11 on bone is mediated by other mechanism(s) than the suppression of Wnt inhibitors. Further details of mechano-sensing mechanism leading to the

enhanced expression of IL-11 as well as the downstream signals of IL-11 leading to enhanced bone formation remain to be clarified.

In conclusion, the present observations using osteoblasts/osteocytes-specific deletion of IL-11 explain how the body can adapt physiologically to exercise-induced mechanical loading by enhancing the expression of IL-11 in the bone that enhances bone formation in one hand and acts as a hormone in the other hand to reduce adipose tissue mass as an energy source via the suppression of adipogenesis. We also clarified a cascade of IL-11 action via IL-11Rα-STAT1/3 activation to enhance Wnt signaling via the suppression of Sost expression in the bone and Dkk1, 2 expression in the adipose tissue (Fig. 8). These findings may provide an important insight for our understanding of osteoporosis and metabolic syndrome.

## Methods

All animal experiments were performed in accordance with the guidelines of the Animal Research Committee, Tokushima University (T2020-110), and the Institutional Animal Care and Use Committee of RIKEN Kobe Branch. This study was approved by the Genetic Modification Experiment Safety Management Committee of Tokushima University (2022-17).

### Generation of IL-11 knockout mice

IL-11 conventional knockout (IL-11⁻/⁻) mice (Accession No. CDB0614K, https://large.riken.jp/distribution/mutant-list.html) were generated by using TT2 ES cells[30] (Supplementary Fig. 10a). IL-11⁻/⁻ female mice were infertile, but appeared normal with normal body length and growth curve compared to their wild-type (WT) littermates (Supplementary Fig. 6a, b). Genomic PCR confirmed that murine *IL-11* gene was present only in WT mice but not in IL-11⁻/⁻ mice (Supplementary Fig. 10b). Female mice were used for all experiments.

### Breeding conditions

C57BL/6 strain female mice were used in this study. All the mice were housed in SPF conditions, 12 h light/dark cycle in 22–25 °C with 60 ± 5% humidity. To determine the effect of high-fat diet (HFD)-induced obesity, mice of each genotype were allocated into two groups with either a regular diet (RD) or a HFD after weaning at 4 weeks with free access to water. Micro-CT analysis of adipose tissue areas was performed at 12 and 24 weeks. The time of each measurement is described in the Figure legends. Mice were anesthetized using isoflurane and euthanized by cervical dislocation at the time of tissue extraction, or when they lost more than 25% of body weight or showing abnormal behavior as approved by the Animal Research Committee of Tokushima University.

Compositions of RD (MFG chow, Oriental Yeast Co. Ltd, Japan) and HFD (F2HFHSD diet, Oriental Yeast Co. Ltd, Japan) are shown in Supplementary Table 1. For HFD, the source of fat was tallow (14 wt%), lard (14 wt%) and soybean oil (2 wt%), the source of carbohydrate was sucrose (20 wt%), cornstarch (14.87 wt%), and the source of protein was casein (25 wt%). Other constituents included cellulose powder

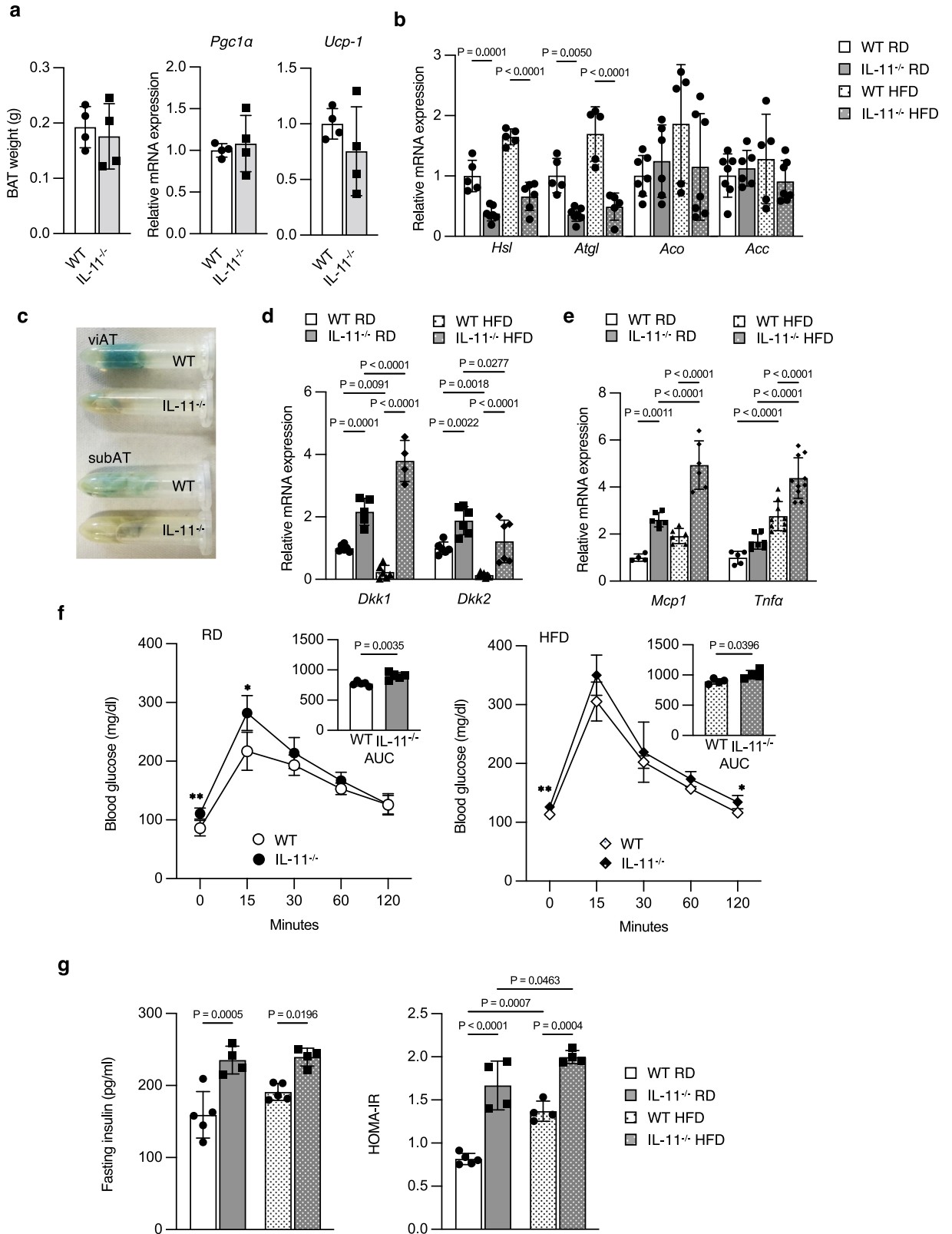

(5 wt%), AIN-93 vitamin mixture (1 wt%) and AIN-93G mineral mixture (3.5 wt%).

For experiments that studied the role of Wnt signal in the development of IL-11 action in the adipose tissue, a β-catenin agonist[22] and a Dkk1 antagonist[23], WAY-262611 (Code No. S9828, Selleck Biotech Co., Tokyo, Japan) was orally given to mice at a concentration of 1 mg/ 1 mL vehicle containing 2% Tween-80 in 0.5% methylcellulose 3 days a week

(4 mg/kg mice/dose on Monday and Wednesday and 6 mg/kg mice/dose on Friday) starting at the age of 4 weeks to the end of experiments at the age of 12 weeks.

**Generation of TOPGAL;IL-11$^{-/-}$ mice**

Wnt indicator TOPGAL mice were obtained from The Jackson Laboratory (Bar Harbor, ME, USA). TOPGAL transgenic mice that express

**Fig. 5 | Increased adiposity with reduced expression of lipolytic genes, enhanced expression of inflammatory cytokines, and glucose intolerance with reduced insulin sensitivity in IL-11$^{-/-}$ mice. a** Brown adipose tissue (BAT) weight and the expression of BAT-specific genes in WT and IL-11$^{-/-}$ mice under RD and HFD. BAT weight and expression of thermogenesis genes, *Pgc1α* and *Ucp-1* in BAT were measured at 12 weeks. *Pgc1α PPAR-gamma coactivator 1 alpah, Ucp-1 uncoupling protein 1.* *n* = 4. Data are means ± SD. No significant difference was observed between WT and IL-11$^{-/-}$ mice by two-tailed Student's unpaired *t*-test. **b** Expression of lipolysis-associated genes, *Hsl, hormone-sensitive lipase* in WAT at 32 weeks of WT (*n* = 5) and of IL-11$^{-/-}$ mice on RD (*n* = 7) or HFD (*n* = 6) and *Atgl, adipose triglyceride lipase* in WAT at 32 weeks of WT (*n* = 5) and of IL-11$^{-/-}$ mice on RD (*n* = 7) or HFD (*n* = 5). Expression of β-oxidation-associated gene *Aco, acyl-CoA oxidation* in WAT at 32 weeks of WT on RD (*n* = 6) or HFD (*n* = 5) and of IL-11$^{-/-}$ mice on RD (*n* = 6) or HFD (*n* = 7), and lipogenesis-associated gene *Acc, acetyl CoA carboxylase* in WAT at 32 weeks of WT on RD (*n* = 5) or HFD (*n* = 7) and of IL-11$^{-/-}$ mice on RD (*n* = 6) or HFD (*n* = 7). Data are means ± SE. *P* values are calculated using two-way ANOVA with Tukey's multiple comparisons test. **c** X-gal staining of visceral (viAT) and subcutaneous AT (subAT) from WT and IL-11$^{-/-}$ mice. **d** Expression of Wnt inhibitors,

*Dkk1* in WAT from WT (*n* = 6) and IL-11$^{-/-}$ mice on RD (*n* = 5) or HFD (*n* = 4), and *Dkk2* in WAT from WT on RD (*n* = 6) or HFD (*n* = 7) and of IL-11$^{-/-}$ mice on RD or HFD (*n* = 6). Data are means ± SD. *P* values are calculated using two-way ANOVA with Tukey's multiple comparisons test. **e** Expression of inflammatory cytokines *MCP1, monocyte chemoattractant protein 1* in WAT from WT on RD (*n* = 4) or HFD (*n* = 6) and IL-11$^{-/-}$ mice on RD or HFD (*n* = 6), and *TNFα, tumor necrosis factor α* in WAT from WT on RD (*n* = 5) or HFD (*n* = 9) and IL-11$^{-/-}$ mice on RD (*n* = 7) or HFD (*n* = 10). Data are means ± SD. *P* values are calculated using two-way ANOVA with Tukey's multiple comparisons test. **f** Changes in blood glucose by oral glucose tolerance test (oGTT) at 24 weeks of WT on RD (*n* = 6) or HFD (*n* = 4) and of IL-11$^{-/-}$ mice on RD (*n* = 5) or HFD (*n* = 4). Data are means ± SD. *P* = 0.0106, **P* = 0.0074 on RD, and *P* = 0.0465, **P* = 0094 on HFD between WT and IL-11$^{-/-}$ mice. *P* values are calculated by two-tailed Student's unpaired *t*-test. Insets show AUC of blood glucose from WT on RD (*n* = 6) or HFD (*n* = 4) and of IL-11$^{-/-}$ mice on RD (*n* = 5) or HFD (*n* = 4). Data are means ± SD. *P* values are calculated by two-tailed Student's unpaired *t*-test. **g** Insulin levels of WT (*n* = 5) and IL-11$^{-/-}$ mice (*n* = 4). HOMA-IR of WT on RD (*n* = 5) or HFD (*n* = 4) and IL-11$^{-/-}$ mice (*n* = 4). Data are means ± SD. *P* values are calculated by two-tailed Student's unpaired *t*-test.

β-galactosidase as a reporter in the presence of TCF/LEF-mediated signaling pathway (Strain #004623, RRID: IMSR_JAX:004623) were bred with IL-11$^{-/-}$ and WT mice to generate TOPGAL;IL-11$^{-/-}$ and TOPGAL;WT mice.

## Generation of conditional IL-11 knockout mice

To establish IL-11 floxed mice (Accession No. CDB1231K: https://large.riken.jp/distribution/mutant-list.html) in which exon 4 of IL-11 gene was flanked by loxP sequences, IL-11-targeted mice carrying a flippase (FLP) recognition target (FRT)-flanked neomycin resistance gene cassette were generated by using TT2 ES cells[30] (Supplementary Fig. 9a). Because these homozygous IL-11-targeted female mice were infertile, we obtained embryo from these mice, and deleted neomycin cassette using FLP-FRT recombination by electroporation of FLP mRNA into the embryo. Genomic PCR confirmed that the size of murine *IL-11* gene was larger in floxed mice compared with WT mice (Supplementary Fig. 9b). Transgenic mice expressing Cre recombinase under the control of osteocalcin promoter (Ocn-Cre)[31] (courtesy of Prof. Hiroshi Takayanagi and Dr. Kazuo Okamoto, University of Tokyo) and adiponectin promoter (Apn-Cre)[32] (courtesy of Prof. Wataru Ogawa and Dr. Tetsuya Hosooka, Kobe University) were mated with IL-11 floxed (IL-11$^{fl/fl}$) mice to obtain conditional knockout mice. Genomic nested PCR of Ocn-Cre;IL-11$^{fl/fl}$, Apn-Cre;IL-11$^{fl/fl}$ and IL-11$^{fl/fl}$ confirmed that exon 4 of IL-11 gene was deleted in the bone only in Ocn-Cre;IL-11$^{fl/fl}$ mice, and was deleted in the AT only in Apn-Cre;IL-11$^{fl/fl}$ mice (Fig. 6a).

## Genotyping

Genomic DNA extracted from mice tails was analyzed by PCR. Genotyping was performed according to the genomic PCR protocol. Briefly, 12.7 μl ddH$_2$O, 2 μl 10× buffer, 1.6 μl dNTP, 0.8 μl 12.5 pmol/l forward and reverse primers, 0.1 μl Ex Taq DNA polymerase (TaKaRa Bio Inc., Japan) and 3 μl DNA sample were used. The PCR program used was as follows: 95 °C for 5 min, followed by 35 cycles of 95 °C for 1 min, 60 °C for 1 min, 72 °C for 1 min, and the final step of 75 °C for 5 min.

Different primer sets were used to distinguish WT and conventional IL-11$^{-/-}$ mice: for WT forward (intron 2) 5′-agattggagggacagggaat-3′, reverse (intron 4) 5′-atttgggggacacaaaacaa-3′, and for IL-11$^{-/-}$ forward (exon 2/intron 2) 5′-agctgcacagatggtaggagattg-3′, reverse (within NeoR cassette) 5′-tatgatcggaattcgatagcggcc-3′. The sizes of the PCR product of WT allele was 737 bp, and that of KO was 401 bp.

In order to distinguish IL-11$^{fl/fl}$ from WT and conditional knockout mice, we first amplified *IL-11* gene with the following primers: forward (intron 2) 5′-ttgggcacttgacgaagggg-3′, reverse (intron 4) 5′-ggcatcttaagacctaggcctc-3′. To distinguish IL-11$^{fl/fl}$, Ocn-Cre;IL-11$^{fl/fl}$, and Apn-Cre;IL-11$^{fl/fl}$, the following primer sets were used for nested PCR:

forward (exon 3) 5′-atgagcgctgggacattggg-3′, reverse (intron 4) 5′-tcatgggctgcgatttgggg-3′. The sizes of the PCR products of IL-11$^{fl/fl}$ and the conditional KO mice, Ocn-Cre;IL-11$^{fl/fl}$ and Apn-Cre;IL-11$^{fl/fl}$, were 890 bp and 254 bp, respectively.

## Establishment of *Il11ra* gene knockout cell lines by VIKING method

CRISPR/Cas9-mediated *Il11ra* gene editing was conducted following the VIKING method described previously[33]. For genome editing of the mouse *Il11ra* gene locus (Ensemble ID: ENSMUST00000098132.10, https://www.uniprot.org/uniprotkb/Q64385/entry), annealed oligonucleotides comprising the sequences of *Il11ra* gene (5′-CACCGAT TCCACCCGCAGTCCTTG-3′, 5′-AAACCAAGGACTGCGGGTGGAATC-3′) were cloned into pX330 (Addgene; #42230) as a locus-specific cleaving vector. For the VIKING method, a donor vector pVKG1-Puro (Addgene; #108670) and a donor cleaving vector VKG1-gRNA-pX330 (Addgene; #108671) were prepared as VIKING modules.

MC3T3-E1 cells (RIKEN Cell Bank, Cat. No. RCB1126) were suspended in Opti-MEM (11058-021; Life Technologies, USA) and transfected with 15 μg DNA in the VIKING modules and *Il11ra* locus-specific cleaving vector using a Lipofectamine LTX reagent (15338100; Life Technologies) according to the manufacturer's protocol. Transfected cells were seeded into 100-mm dishes and pre-cultured in MEMα without antibiotics for 24 h. C3H10T1/2 cells (RIKEN Cell Bank, Cat. No. RCB0247) were suspended in Opti-MEM and transfected with 15 μg DNA in the VIKING modules and *Il11ra* locus-specific cleaving vector using a Lipofectamine 2000 reagent (11668019; Life Technologies) according to the manufacturer's protocol. Transfected cells were seeded into 100-mm dishes and pre-cultured in DMEM without antibiotics for 24 h. Transduced MC3T3-E1 or C3H10T1/2 cells were selected following puromycin treatment (0.1 μg/mL) for 14 days to isolate clonal colonies.

For genome sequencing, PCR amplification from genomic DNA of each isolated cell line was conducted using primers "5′-TCGT CGGCAGCGTCAGATGTGTATAAGAGACAGACACACACTGTGGGAAGG AAT-3′" and "5′-GTCTCGTGGGCTCGGAGATGTGTATAAGAGACAGG ACCGAGACACACTGCAGAC-3′" with GoTaq master mix (M7132; Promega, USA) or KOD FX Neo (KFX-201; Toyobo, Japan) according to the manufacturer's protocol. Each PCR product was directly sequenced using next-generation sequencing (MiSeq; Illumina, USA) (Supplementary Fig. 1a−c).

## Immunoblotting

Cells were collected and lysed in lysis buffer (Thermo Fisher Scientific) after stimulation of mIL-11 (R&D systems), electrophoresed on a

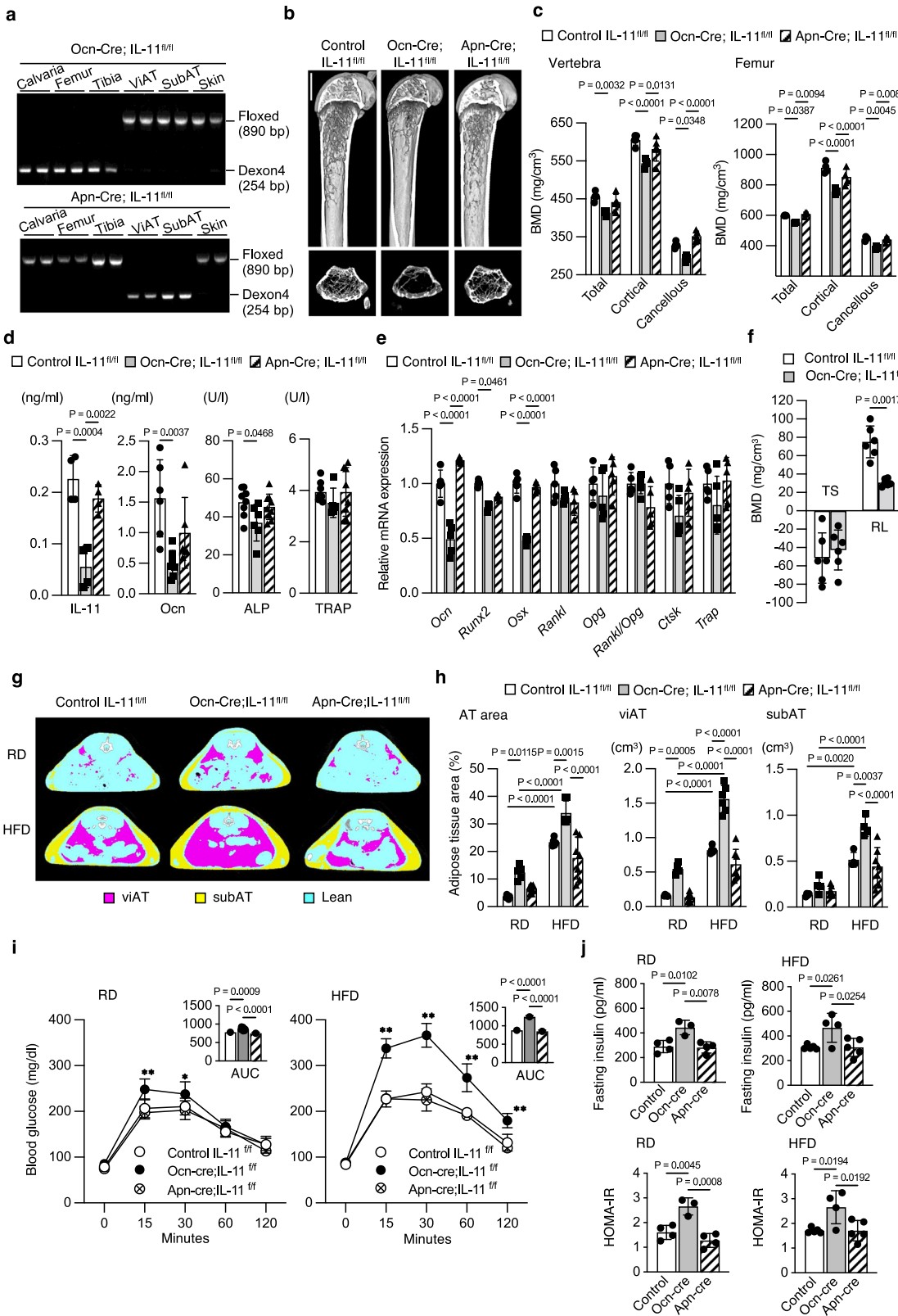

10% SDS-PAGE, and blotted onto PVDF membranes (Biorad). After blocking, the membranes were incubated with primary antibodies overnight at 4 °C, and then with HRP-conjugated secondary antibodies for 1 h. Protein bands were visualized with a SuperSignal West Pico Chemiluminescent Substrate (Thermo Fisher). The sources for the antibodies were as follows: Rabbit monoclonal antibodies to STAT1, STAT3, p-STAT1, HDAC4, HDAC5, GAPDH (dilution 1:1000).

Rabbit polyclonal antibody to p-STAT3 (1:1,000), mouse monoclonal antibody to β-actin (1:1000), goat anti-rabbit IgG HRP-linked antibody, and goat anti-mouse IgG HRP-linked antibody (dilution 1:10,000) are from Cell Signaling (Danvers, MA). Mouse monoclonal antibody to NMP84 (dilution 1:1000) as a nuclear marker was obtained from Abcam, and STAT3-IN-1 from Selleck Biotech Co., Tokyo, Japan.

**Fig. 6 | Bone and adipose tissue phenotypes of osteoblast/osteocyte-specific IL-11 deleted mice recapitulate systemic IL-11[−/−] mice. a** PCR analysis of genomic DNA extracted from calvaria, femur, tibia viAT, suAT and skin of Ocn-Cre;IL-11[fl/fl] and Apn-Cre;IL-11[fl/fl] mice. Representative figures of three independent experiments. **b** Micro-CT of femoral bones from 12-week-old IL-11[fl/fl] control, Ocn-Cre;IL-11[fl/fl] and Apn-Cre;IL-11[fl/fl] mice. Scale bar = 1 mm. **c** Total, cortical and cancellous BMD at 12 weeks of vertebral bone from control, Ocn-Cre;IL-11[fl/fl] and Apn-Cre;IL-11[fl/fl] mice ($n = 5$), and femoral bone from control, Ocn-Cre;IL-11[fl/fl] and Apn-Cre;IL-11[fl/fl] mice ($n = 4$). Data are means ± SD. *P* values are calculated using two-way ANOVA with Tukey's multiple comparisons test. **d** Serum IL-11 ($n = 4$) and bone turnover markers, Ocn, osteocalcin in control ($n = 6$), Ocn-Cre;IL-11[fl/fl] ($n = 7$) and Apn-Cre;IL-11[fl/fl] mice ($n = 6$), ALP, alkaline pohosphatase in control ($n = 8$), Ocn-Cre;IL-11[fl/fl] ($n = 6$) and Apn-Cre;IL-11[fl/fl] mice ($n = 9$), and TRAP, tartrate-resistant acid phosphatase in control ($n = 8$), Ocn-Cre;IL-11[fl/fl] ($n = 5$) and Apn-Cre;IL-11[fl/fl] mice ($n = 8$). Data are means ± SD. *P* values are calculated using ordinary one-way ANOVA with Tukey's multiple comparisons test. **e** Expression of osteoblastic and osteoclastic genes in femurs of 12-week-old control, Ocn-Cre;IL-11[fl/fl] and Apn-Cre;IL-11[fl/fl] mice. $n = 5$. Data are means ± SD. *P* values are calculated using two-way ANOVA with Tukey's multiple comparisons test. **f** BMD change after TS and RL in control (open bar) and Ocn-Cre;IL-11[fl/fl] mice (closed bar). $n = 6$. Data are means ± SD. *P* values are calculated using two-way ANOVA with Bonferroni's multiple comparisons test. **g** Micro-CT scan at L5 level and **h** quantitative analysis at 12 weeks of total AT area in control on RD ($n = 6$) or HFD ($n = 5$), Ocn-Cre;IL-11[fl/fl] on RD ($n = 5$) or HFD ($n = 6$) and Apn-Cre;IL-11[fl/fl] mice on RD ($n = 8$) or HFD ($n = 7$), of viAT (visceral AT) area in IL-11[fl/fl] control on RD or HFD ($n = 5$), Ocn-Cre;IL-11[fl/fl] on RD or HFD ($n = 6$) and Apn-Cre;IL-11[fl/fl] mice on RD or HFD ($n = 7$), and of subcutaneous AT area in control on RD or HFD ($n = 4$), Ocn-Cre;IL-11[fl/fl] on RD or HFD ($n = 4$) and Apn-Cre;IL-11[fl/fl] mice on RD or HFD ($n = 7$). Pink, viAT; yellow, subAT; blue, lean mass; white, vertebral bone. AT area at 12 weeks. Data are means ± SD. *P* values are calculated using two-way ANOVA with Tukey's multiple comparisons test. **i** Changes in blood glucose by oral glucose tolerance test (oGTT) at 24 weeks of IL-11[fl/fl] control ($n = 8$), Ocn-Cre;IL-11[fl/fl] ($n = 8$) and Apn-Cre;IL-11[fl/fl] mice ($n = 6$) on RD or HFD. Data are means ± SD. *$P = 0.0039$, **$P < 0.0001$ between control and Ocn-Cre;IL-11[fl/fl] or Apn-Cre;IL-11[fl/fl] mice using two-way ANOVA with Tukey's multiple comparisons test. Insets show AUC of blood glucose. Data are means ± SD. *P* values are calculated using ordinary one-way ANOVA with Tukey's multiple comparisons test. **j** Insulin levels and HOMA-IR of control on RD ($n = 4$) or HFD ($n = 5$), Ocn-Cre;IL-11[fl/fl] on RD ($n = 3$) or HFD ($n = 4$) and Apn-Cre;IL-11[fl/fl] mice on RD ($n = 4$) or HFD ($n = 5$). Data are means ± SD. *P* values are calculated using ordinary one-way ANOVA with Tukey's multiple comparisons test.

## Tail suspension model

Tail suspension was performed as previously described[4]. Female 8-week-old WT and IL-11[−/−] mice were used. In brief, tails of the mice were suspended to lift off the hind limbs from ground for two weeks. During the period of tail suspension, mice were able to access food and water freely. After the tail suspension, mice were allowed to move freely in the ground for three weeks (Reloading group) or sacrificed (Tail suspension group) to obtain serum and bone samples.

## X-gal staining

β-Galactosidase staining of bone tissues and adipocytes was conducted as previously described[34]. Briefly, tissues were collected and fixed in 2% (v/v) paraformaldehyde and 0.02% (v/v) glutaraldehyde for 1 h, and bone tissues were decalcified in 15% EDTA for 10 days, and then incubated in 0.1% 4-chloro-5-bromo-3-indolyl β-D-galactosidase (X-Gal solution; Wako, Osaka, Japan) at 37 °C overnight. Tissues were washed, then followed by post-fixation with 4% paraformaldehyde at 4 °C for 16 h, and rinsed in an ascending series of ethanol, embedded in paraffin, sectioned and counterstained with eosin.

## Immunostaining

Tibias were excised and subsequently fixed in 4% paraformaldehyde at 4 °C overnight, then decalcified in 15% EDTA for 15 days. Samples were dehydrated in ethanol, embedded in paraffin blocks, and then cut into 10 μm sections. Before staining, the sections were deparaffinized in xylene, and pretreated in 3% (v/v) hydrogen peroxide/methanol (Wako, Osaka, Japan), followed by blocking using Protein Block-Serum Free (Dako, CA, USA). Then slides were incubated in 1.5% normal goat serum for 30 min at room temperature using Vectastain ABC Goat IgG kit (Vector Laboratories Inc., CA, USA). The primary antibody (Goat anti-sclerostin, Goat anti-Dkk1, and Goat anti-Dkk2, R&D Systems, Minneapolis, MN, USA) was 1:100 diluted and incubated at 4 °C overnight. Slides were washed and incubated in secondary antibody (Vectastain kit) diluted 1:200 for 30 min at room temperature, followed by ABC reagent using Vectastain kit for 30 min in accordance to the manufacture's protocol. After washing, slides were developed in a working solution of Imm PACT™ DAB Peroxidase Substrate kit (Vector Laboratories Inc., Burlingame, CA, USA), followed by counterstaining with Weak Methyl Green (Dako, CA, USA).

## Real-time PCR analysis

Total RNA was extracted from femoral bones or adipose tissues and isolated with TRIzol reagent (Invitrogen, CA, USA) according to the manufacture's protocol. Complementary DNA was synthesized using PrimeScript RT Master Mix (Takara Bio Inc., Japan). Then samples were subjected to quantitative real-time PCR analysis using ABI 7300 Real-Time PCR system (Applied biosystems, Foster City, CA) with SYBR® Green Premix Ex Taq™II kit (Takara, Shiga, Japan). The sequences of primers are listed in Supplementary Table 2.

## ELISA

To determine the serum bone metabolic parameters, Mouse Osteocalcin EIA Kit (Biomedical Technologies Inc., MA, USA), Mouse TRAP Assay (Immuno diagnostic system Ltd, UK) and Alkaline phosphatase (ALP) test kit (Wako, Osaka, Japan) were used for measuring serum samples in accordance with the manufactures' protocols.

Serum leptin, adiponectin and insulin concentrations were measured with Mouse and Rat Leptin ELISA kit (BioVendor, Brno, Czech Republic), Mouse Adiponection ELISA kit (BioVendor, Brno, Czech Republic) and Ultra Sensitive Mouse Insulin ELISA kit (Morinaga Institute of Biological Science, Inc., Yokohama, Japan), respectively, according to the manufactures' protocols.

## Micro-computed tomography (μCT) analysis

Before and during μCT scan, mice were anesthetized by inhalation of isoflurane (Abbott, Tokyo, Japan). Mice were placed on abdominal position in 48 mm wide specimen holder with 96 mm pixel resolution. Hindlimbs were extended to keep the femur and spine into right angle, and then scanned from the proximal end of L1 vertebra to the distal end of L5 to measure vertebral BMD, and whole femoral bones were scanned using LaTheta LCT-200 (Hitachi-Aloka, Tokyo, Japan), as previously described[35]. Calvaria bone samples were extracted and stored. Bone mineral density (mg/cm³) and adipose tissue area were calculated by LaTheta software.

## Bone histomorphometric analysis

Mice were double-labeled with 16 mg/kgBW calcein (Sigma, St. Louis, USA) at 6 and 2 days before sacrifice. Lumbar vertebra were removed and fixed in 4% paraformaldehyde (PFA) at 4 °C overnight, followed by dehydration with a series of ethanol, then embedded in methyl methacrylate monomer (MMA, Wako, Japan). The plastic sections were cut by a standard microtome (LEICA) into 7 μm samples for von Kossa staining and 4 μm for TRAP and Toluidine blue staining. Histomorphometric analysis was performed by OsteoMeasure (OsteoMetrics, Inc.,GA, USA) according to the ASBMR guideline[36].

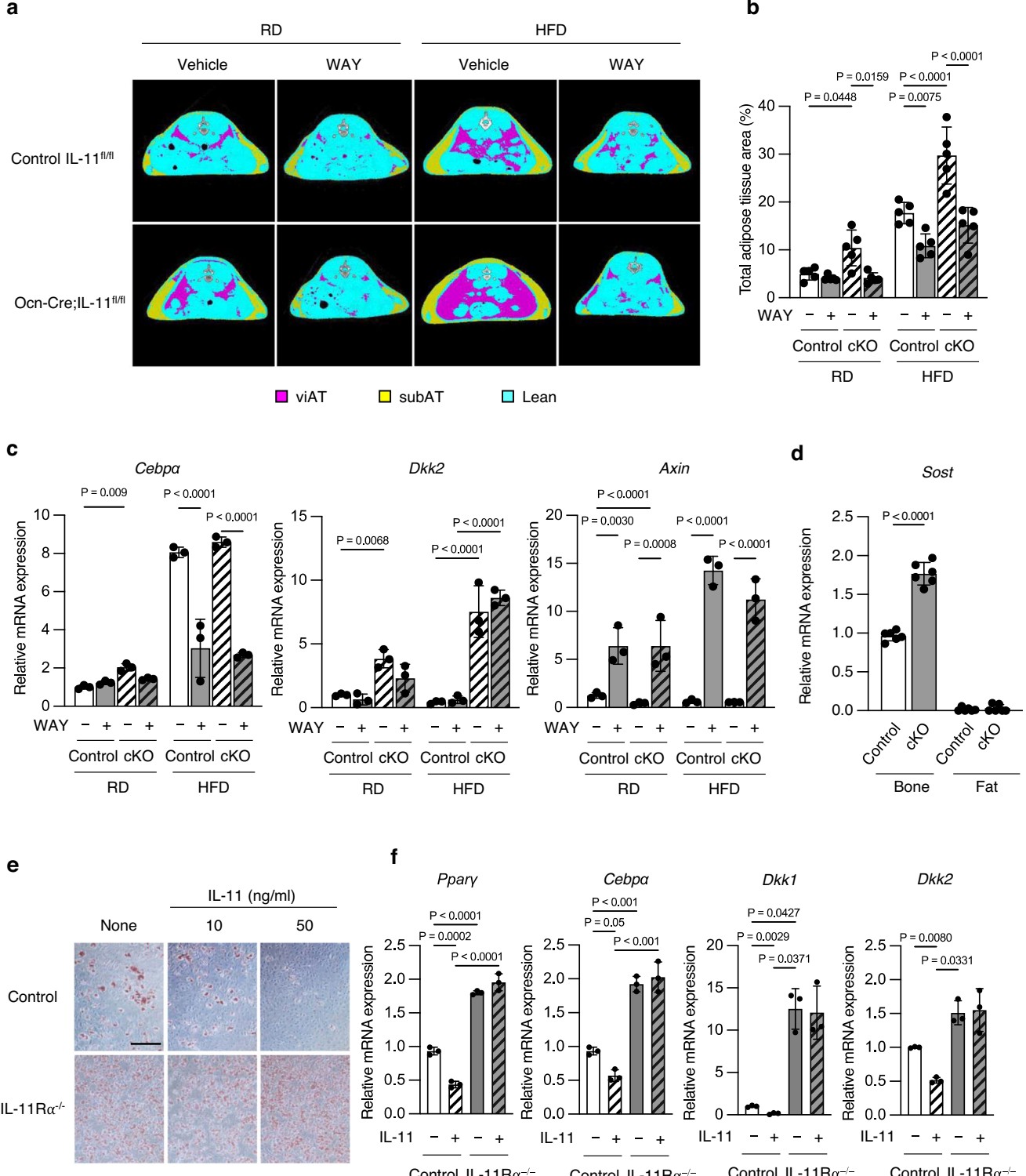

**Fig. 7 | Signaling pathway of IL-11 in the adipose tissue in osteoblast/osteocyte-specific IL-11$^{-/-}$ mice. a** Micro-CT scan at L5 level. Pink, viAT; yellow, subAT; blue, lean mass; white, vertebral bone, and **b** quantitative analysis of adipose tissue in IL-11$^{fl/fl}$ control and Ocn-Cre;IL-11$^{fl/fl}$ mice on RD or HFD in the presence or absence of WAY-262611, a β-catenin agonist and a Dkk1 antagonist. WAY-262611 at a dose of 2 mg/kg/day was orally given to mice 3 days a week (4 mg/kg mice/dose on Monday and Wednesday and 6 mg/kg mice/dose on Friday) starting at the age of 4 weeks to the end of experiments at the age of 12 weeks. $n = 5$. Data are means ± SD. *P* values are ca+culated using two-way ANOVA with Tukey's multiple comparisons test. **c** Expression of an adipogenic gene, *Cebpα, CCAAT enhancer binding protein α*, a Wnt inhibitor, *Dkk2, dickopf2*, and a Wnt target gene, *Axin*, in the adipose tissue of IL-11$^{fl/fl}$ control and Ocn-Cre;IL-11$^{fl/fl}$ mice on RD or HFD in the presence or absence of WAY-262611 at the age of 12 weeks. $n = 3$. Data are means ± SD. *P* values are calculated using two-way ANOVA with Tukey's multiple comparisons test. **d** Expression of *Sost* in the bone and the adipose tissue in IL-11$^{fl/fl}$ control and Ocn-Cre;IL-11$^{fl/fl}$ mice. $n = 6$. Data are means ± SD. *P* values are calculated by two-way ANOVA with Sidak's multiple comparisons test. **e** Oil-Red O staining of WT and IL-11Rα$^{-/-}$ C3H10T1/2 cells in an adipogenic medium and cultured for 7 days. Representative pictures of three independent experiments. Scale bar = 100 μm. **f** Expression of adipogenic genes, *Pparγ and Cebpα*, and Wnt inhibitor genes, *Dkk1, 2*, in WT and IL-11Rα$^{-/-}$ C3H10T1/2 cells in an adipogenic medium and cultured for 7 days in the presence or absence of 10 ng/mL IL-11. $n = 3$. Data are means ± SD. *P* values are calculated using ordinary one-way ANOVA with Tukey's multiple comparisons test.

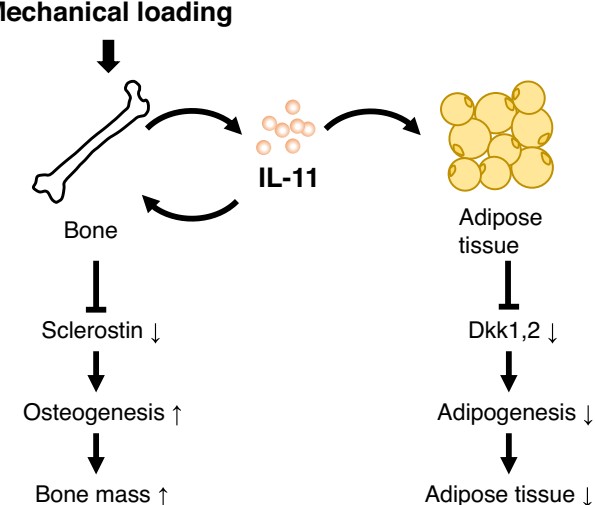

**Fig. 8 | Diagram of the summary of the present observations in the regulation of bone and adipose tissue mass by exercise-induced mechanical loading to the bone.** Exercise creates mechanical loading to the bone, which stimulates IL-11 expression in the bone. IL-11 produced in the bone acts locally to enhance osteogenesis via a stimulation of Wnt signaling by suppressing the expression of Wnt inhibitors including sclerostin, and increases bone mass. At the same time, exercise-induced increase in IL-11 in the bone acts like a hormone, and suppresses the expression of Wnt inhibitors, Dkk1 and 2, in the adipose tissue. The enhanced Wnt signaling in the adipose tissue suppresses adipogenesis to increase energy supply for exercise.

## Ex vivo cell culture
Bone marrow cells were extracted from 12-week-old female mice. For osteoblastic differentiation, bone marrow stromal cells (BMSC) were cultured in α-Modified Eagle's minimal essential medium (α-MEM; Gibco, NY, USA) supplemented with 10% fetal bovine serum (FBS, Thermo, Utah, USA) and induced with 50 μg/ml ascorbic acid (Wako) and 10 mM β-glycerophosphate (Sigma-Aldrich, Tokyo, Japan) (osteoblast differentiation medium). For adipogenic differentiation, BMSC were incubated with $10^{-6}$M Troglitazone (Sigma-Aldrich, Tokyo, Japan).

## Oil-Red O staining
For Oil-Red O staining, BMSC were cultured for 14 days, fixed and rinsed in 60% 2-propanol for 3 times, then stained with Oil-Red O staining solution (Sigma-Aldrich, Tokyo, Japan) for 30 min at room temperature. The cells were again washed and rinsed with 60% 2-propanol. The Oil-Red O positive cell number was counted under microscope.

## Alkaline phosphatase (ALP) staining and Alizarin Red staining
For ALP staining, cells were cultured in osteoblast differentiation medium for 7 days and fixed in 3.7% formaldehyde for 10 min. Then cells were incubated at 37 °C with freshly prepared 1 mg/ml Naphthol-AS phosphatase (Wako), 6 mg/ml Fast-Blue BB (Wako), 0.5% (v/v) N,N-dimetylformaide, 1 mM MgCl₂, 1 M Tris-HCl (pH = 8.8) and stained for 5 min. Cells were cultured in osteoblast differentiation medium for 15 days for Alizarin Red staining, fixed in 10% formaldehyde for 10 min, washed and stained with 0.2% Alizarin Red (Sigma)/1 M Tris-HCl (pH = 8.3) at 37 °C for 20 min.

## Determination of adipocyte size and number
Visceral fat pad was extracted to determine the adipocyte size and number as previously described[37]. In brief, WAT was fixed with osmium tetroxide (Sigma-Aldrich, Tokyo, Japan) and suspended in isotonic saline. To remove fibrous elements and trap adipocytes, samples were passed through 250 μm and 25 μm nylon filters, respectively. Adipocyte size was analyzed using a Coulter counter equipped with a 560 μm aperture tube and a multichannel particle analyzer (Multisizer II, Coulter Electronics, Fullerton, CA). Adipocyte relative number was determined by dividing the total WAT weight (mg) by the estimated mean adipocyte weight (mg), which was calculated by adipocyte density (0.948 mg/ml) × mean adipocyte volume (the average value of adipocyte diameter).

## Glucose tolerance test
Mice were fasted for 16 h before oGTT and then administrated with 1 g/kgBW glucose orally. Blood samples were collected from mice tails at various time points, and blood glucose was measured by glucometer (Medisafe mini GR-102, Terumo, Tokyo, Japan). HOMA-IR (homeostatic model assessment insulin resistance) as an index of insulin sensitivity was calculated by the following formula, where 100 pg/mL insulin corresponds to 2.6 μU/mL:

Fasting serum insulin (μU/mL) × fasting blood glucose (mg/dL)/ 405 = HOMA-IR

## Statistical analysis
Information on biological replicates (n) is indicated in the figure legends. All statistical analyses were performed by GraphPad Prism Version 9.4.1. For the in vivo experiments, sample sizes were determined on the basis of previous publications of similar studies and previous experiments. The statistical tests used in each panel are mentioned in the figure legends. Results were considered significant when $P$ values are below 0.05.

## Reporting summary
Further information on research design is available in the Nature Portfolio Reporting Summary linked to this article.

## Data availability
The data supporting the findings from this study are available within the manuscript and its supplementary information. Source data are provided with this paper.

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

## Acknowledgements

This study was supported in part by Grants-in-Aid for Scientific Research (#20249050, #25293215 and #16H05327 to T.M.; #17H05104 and #19K22719 to M.H.; #18K19518 to S.S.; and #19H03676 to S.F.) from the Japan Society for the Promotion of Science.

## Author contributions

T.M., M.A. and S.F. designed the study. B.D., M.H., I.E., T.N., S.K., M.A. and T.M. analyzed the data. B.D., M.H., I.E. and T.M. conducted the experiments. I.E., Y.H., Y.O., T.K., Y.T. and M.T. acquired the data on bone phenotype. R.K., S.S. created and provided cultured cell lines, and acquired data. S.S., H.K. and G.S. created and provided IL-11 gene deleted and floxed mice. B.D., M.H., Y.H., Y.T., M.T., and H.S. conducted the experiments on adipose tissue phenotype and glucose metabolism, and analyzed those data. T.N., S.K. and T.M. wrote the manuscript. B.D. and M.H. contributed equally to this work. B.D. thoroughly examined systemic IL-11 deleted mice and originally found the increase in adipose tissue mass. MH followed the study, examined conditional IL-11 deleted mice, and found the importance of IL-11 actions in osteoblasts/osteocytes. Therefore, BD is listed first All the authors meet the ICMJE authorship requirements.

## Competing interests

The authors declare no competing interests.
