## [Peer Review File · Nature Communications]

Osteoblast/osteocyte-derived interleukin-11 regulates osteogenesis and systemic adipogenesisREVIEWER COMMENTS

Reviewer #1 (Remarks to the Author):

The manuscript „Osteoblast/osteocyte-derived interleukin-11 regulates osteogenesis and systemic adipogenesis” by Bingzi Dong and colleagues identifies the cytokine interleukin-11 (IL-11) as a factor that is secreted from bones that communicates with adipose tissue. They show that systemic IL-11 deletion results in reduced bone mass, suppressed bone formation, enhanced expression of Wnt inhibitors, suppressed Wnt signaling, increased systemic adiposity and glucose intolerance. Using floxed IL-11 mice, the authors further demonstrate that deletion of IL-11 in Osteoblasts/osteocytes recapitulates the findings of the total IL-11 KO mice, whereas deletion of IL-11 in adipocytes has no effect.

Inter-organ cross-talk is an exciting topic, and the identification of a soluble factor that is released from bone cells and acts on metabolic phenotypes is exciting. IL-11 has gained attention in recent years due to its involvement in different human inflammatory diseases, but is long known to be involved in bone homeostasis. The link to obesity and insulin resistance, however, is new and has not been explored before. This is an important finding that deserves publication in journal like Nat Comm.

I have several comments related to different parts of the manuscript that the authors should address when revising their manuscript.

General comments:

- The most closely related cytokine to IL-11 is IL-6, which has also an important role in bone homeostasis and obesity/metabolism. This should be mentioned in the manuscript. It would not be surprising if IL-6 levels are increased in the IL-11 KO mice to compensate for the loss of IL-11. It should be tested whether IL-6 is increased in the IL-11 KO mice compared to WT littermates.
- The authors focus on proteins involved in wnt signaling as executing molecules downstream of IL-11 signaling. However, this connection is not well established in the literature and also not convincingly demonstrated in the manuscript. When cells are stimulated by a cytokine, thousands of genes are up- or down-regulated. I have no doubt that genes involved in wnt signaling are among them as demonstrated by the authors, but they have to show a mechanistic link to IL-11 signaling. At the moment, the data only show a correlation; the authors have to show a causative relationship between IL-11 and wnt signaling.

Further specific points:

- Introduction: The authors summarize the role of IL-11 in bone, but do not mention that IL-11 signaling is required for skull formation in mice and also in humans (summarized in PMID: 33940443). This could be added and discussed, as it links IL-11 deficiency to a human syndrome.
- Fig 1a: There is a scale bar shown, but its length is not defined in the figure legend.
- Fig 1b,c: The figure legend states that “means \pm SD” is shown here, but only one SD is shown for each genotype.
- Fig 2a: TRAP qPCR levels show high variability, and because of this it is unclear whether it is reduced in IL-11 KO mice. Please repeat the experiment to get a clearer picture.
- Fig 2c: From the text, it is not clear for me why the authors focused on Wnt signaling. Is this downstream of gp130/Jak/STAT signaling? The connection to IL-11 has to be better presented.
- Fig 2e: There is a scale bar shown, but its length is not defined in the figure legend. Furthermore, these pictures have to be quantified (stained cells per field) from an appropriate number of images per mouse and from an appropriate number of animals.
- Fig 4a: The authors show pictures from a single time point, and the effect looks striking; however, it would be more informative to show sections from mice at different ages.
- Line 295-299: “Although reasons for the discrepancy between the systemic IL-11^{-/-} and IL-11R α ^{-/-} mice in bone resorptive activity are not clear, there is a possibility that differences in downstream signaling pathways may exist, as IL-11R α mediates the other signaling pathways than the ones activated by IL-11.” This statement is confusing. IL-11 binds to the IL-11R, which induces gp130 homodimerization and the subsequent activation of intracellular signaling cascades. How should IL-11 and IL-11R induce different signaling pathways? Neither is the IL-11R biologically active without IL-11, nor can IL-11 activate gp130 without IL-11R.
- Methods: The authors state “IL-11^{-/-} mice appeared normal with similar fertilizing ability (lines 351/352) and “Because these homozygous IL-11-targeted female mice were infertile (lines 390/391).” Could the authors please comment why these two mouse strains behave differently? IL-11R KO mice have been reported as infertile (PMID: 9679067), so it is interesting to compare the IL-11 KO mice to the IL-11R KO.
- Methods: The authors show the targeting strategy of the two new IL-11 KO strains in the supplement and the results of the PCR are convincing. What is missing is an actual confirmation on the protein level that these mice do not produce IL-11. This could for example be done by stimulating primary fibroblasts from IL-11 KO and WT littermates with TGF beta and monitoring of IL-11 production.

Reviewer #2 (Remarks to the Author):

In this manuscript, the authors report that conventional deletion of IL11 in mice leads to decreased bone mass in two different bones of the skeleton as well as a systemic increase in adiposity and insulin resistance. The decrease in bone density affects both trabecular and cortical bone and is mainly due to decreased bone formation and no significant changes in bone resorption are noted by histomorphometry. The increase in adiposity occurs at multiple site, including the bone marrow, is characterized by an increase in both the number and the size of adipocytes. To determine whether these two phenotypes are independent or linked, the authors then deleted IL11 in the osteoblast lineage using osteocalcin-Cre and in the adipocyte lineage using adiponectin-Cre. Most interestingly, the deletion in osteoblasts recapitulated the entire phenotype whereas the deletion in adipocytes had no measurable effects. PCR analysis of gene expression and genetic analysis of Wnt signaling in vitro and in vivo showed that in all circumstances several Wnt inhibitors were increased in the absence of IL11 in both bone and fat. The authors conclude that IL11 secretion from bone (osteoblasts) acts both as a cytokine, affecting positively bone formation locally, and as a hormone to systemically repress adipogenesis, improve fat metabolism and insulin tolerance. Given that the authors have previously shown that exercise and skeletal loading increase IL11 production in bone, they infer that IL11 from bone is a mechanism by which exercise strengthens bone and increase energy expenditure.

Critique: Overall, this is a very interesting manuscript reporting novel findings of significant importance in the field of bone, fat and glucose metabolism. The main strength is the identification of IL11 as a potential "hormone" secreted by bone to affect systemically fat and metabolism. In fact, the contrast between the Ocn-cre and the Adipoq-cre phenotypes is very striking and, together with the observed changes in serum levels of IL11, supports well their conclusion. There are however a few weaknesses that, if addressed, would make the manuscript much stronger. In particular:

1- The role of Wnt signaling inhibition in IL11-KO mice should be better established, particularly in vivo, possibly by blocking sclerostin and/or Dkk1-2 with antibodies and determining whether the fat and metabolic phenotypes are corrected. Is Wnt inhibited in adipocytes? Since Wnt are short range molecules, if the changes in fat are Wnt-dependent, it implies that IL11 regulates Wnt signaling locally, in these cells.

3- What evidence do we have that adipocytes express the IL11 receptor and respond to IL11? A nice experiment would be to delete the receptor with Adipoq-cre and determine whether this generates the fat and metabolism phenotype in absence of any bone phenotype. Also, there is no data testing the response of adipocytes to IL11, only deletion is tested.

Specific points:

line 142: It is surprising that MC3T3-E1 cells express measurable levels of Soft RNA. This is supposed to define the osteocyte phenotype. How does that compare with osteocyte lines? What number of cycles are required to detect expression?

Fig 2e: the legend does not match

Fig 3: is fat affected by unloading ? is this change in fat, if any, affected by IL11 deletion?

Reviewer #3 (Remarks to the Author):

This study identified bone-derived IL-11 as mechano-sensing factor that regulating bone and fat metabolism. Mice with deletion of IL-11 in osteoblasts displayed reduced bone mass with suppressed bone formation and increased systemic adiposity leading to insulin resistance. Furthermore, results showed that the effect of IL-11 on osteoblastogenesis and adipogenesis in IL-11 KO mice mediated via upregulating the Wnt signaling inhibitors. On the other hand, bone formation and adipose tissue mass was not affected in adipocyte-specific IL-11 knockout mice. This study well conducted, but still descriptive and lacking strong supportive data for the mechanism underplaying the function of IL-11 in bone and fat metabolism.

- Does IL-11 acts on adipocyte cells via IL-11Ra ???. More in vitro mechanistic studies needed in adipocyte cell model to identify the direct / indirect effect of IL-11 on adipogenesis. Such studies should investigate the paracrine mode of IL-11 action on adipogenesis.

- The authors are asked to study the phenotype of newly generated adipocyte specific IL-11Ra KO mice

- Figure S2, since IL-11 is mainly expressed in bone, what is the serum level of IL-11 in osteocalcin-Cre;IL-11fl/fl mice and adiponectin-Cre; IL-11fl/fl mice ?. These data should be added to clarify that the serum of IL-11 in Adipo-IL-11fl/fl mice still high and therefore the phenotype of adipo-IL-11fl/fl mice is not reflecting IL-11 deletion in adipocyte.

-The authors should clarify how their work could provide a novel therapeutic approach to osteoporosis and metabolic syndrome, supplemented with illustrated diagram.

- Is there any clinical relevance of IL-11 to human bone loss related disease. This study should be supported by clinical data to correlate serum level of IL-11 to bone mass, bone markers, lipid metabolism parameters,etc.

Responses to Reviewer Comments

Reviewer #1 (Remarks to the Author):

The manuscript „Osteoblast/osteocyte-derived interleukin-11 regulates osteogenesis and systemic adipogenesis” by Bingzi Dong and colleagues identifies the cytokine interleukin-11 (IL-11) as a factor that is secreted from bones that communicates with adipose tissue. They show that systemic IL-11 deletion results in reduced bone mass, suppressed bone formation, enhanced expression of Wnt inhibitors, suppressed Wnt signaling, increased systemic adiposity and glucose intolerance. Using floxed IL-11 mice, the authors further demonstrate that deletion of IL-11 in Osteoblasts/osteocytes recapitulates the findings of the total IL-11 KO mice, whereas deletion of IL-11 in adipocytes has no effect.

Inter-organ cross-talk is an exciting topic, and the identification of a soluble factor that is released from bone cells and acts on metabolic phenotypes is exciting. IL-11 has gained attention in recent years due to its involvement in different human inflammatory diseases, but is long known to be involved in bone homeostasis. The link to obesity and insulin resistance, however, is new and has not been explored before. This is an important finding that deserves publication in journal like Nat Comm.

I have several comments related to different parts of the manuscript that the authors should address when revising their manuscript.

General comments:

1-1· The most closely related cytokine to IL-11 is IL-6, which has also an important role in bone homeostasis and obesity/metabolism. This should be mentioned in the manuscript. It would not be surprising if IL-6 levels are increased in the IL-11 KO mice to compensate for the loss of IL-11. It should be tested whether IL-6 is increased in the IL-11 KO mice compared to WT littermates.

We thank the Reviewer for raising an important issue about the level of IL-6.

We measured serum IL-6 concentration in WT and IL-11^{-/-} mice. There was no significant difference in serum IL-6 levels between those mice. These results are presented in Figure 2b and explained in the text.

Results, p 7, lines 8 to 10 of the 2nd para: “Although the most closely related cytokine to IL-11 and has also an important role in bone metabolism is IL-6, serum IL-6 concentration in IL-11^{-/-} mice was not different from that in WT mice (Fig 2b).”

1-2- The authors focus on proteins involved in wnt signaling as executing molecules downstream of IL-11 signaling. However, this connection is not well established in the literature and also not convincingly demonstrated in the manuscript. When cells are stimulated by a cytokine, thousands of genes are up- or down-regulated. I have no doubt that genes involved in wnt signaling are among them as demonstrated by the authors, but they have to show a mechanistic link to IL-11 signaling. At the moment, the data only show a correlation; the authors have to show a causative relationship between IL-11 and wnt signaling.

Reasons why we focused on Wnt pathway as a downstream signal was because we already found that mechanical loading enhanced IL-11 expression which suppressed Sost gene expression and enhanced Wnt signaling (ref. 4), and because Wnt pathway is one of the most important signals for the enhancement of bone formation. Nevertheless, we agree with the Reviewer that we need to demonstrate a mechanistic link between IL-11 and Wnt signaling.

In order to show a causative link between IL-11 and Wnt signaling, we asked the following questions: 1) Does IL-11 enhance nuclear translocation of HDAC4/5 in osteoblastic cells, because an enhancement of HDAC4/5 translocation into the nucleus is proposed to be one of the mechanisms whereby Sost gene expression is suppressed by stimulation of bone formation, and 2) is the effect of IL-11 on nuclear translocation of HDAC4/5 mediated via STAT phosphorylation and abolished by inhibition of STAT signal? In order to answer those questions, we examined whether IL-11 enhances nuclear translocation of HDAC4/5 and suppresses Sost gene expression in the presence or absence of a STAT3 inhibitor. We were able to demonstrate that IL-11 stimulated nuclear translocation of HDAC4/5 and suppressed Sost expression, both of which was abrogated in the presence of a STAT3 inhibitor. We presented those data in Figures 2h to 2j, and added the following sentences to explain those data:

Results, p.8, line 9 from the bottom to p.5, line 4: “Because STAT3 is reported to play an important role in regulating osteogenesis²⁰, and because class IIa histone deacetylases, HDAC4 and HDAC5, are shown to be required for loading-induced *Sost* suppression and bone formation²¹, we next examined whether nuclear translocation of HDAC4/5 is enhanced by IL-11 and whether STAT3 inhibition influences nuclear translocation of HDAC4/5 and *Sost* expression in MC3T3-E1 cells. As shown in **Fig. 2h** and **2i**, IL-11 enhanced nuclear translocation of HDAC4/5. A STAT3 inhibitor, STAT3-IN-1, abrogated IL-11-induced nuclear translocation of HDAC4/5. In parallel with the inhibition of HDAC4/5 translocation, inhibition of STAT3 by STAT3-IN-1 abrogated the suppression of *Sost* expression by IL-11 (**Fig 2j**). These results demonstrate that IL-11 acts on osteoblastic cells via IL-11R α to enhance STAT1/3 phosphorylation, which increases nuclear translocation of HDAC4/5 and suppresses *Sost* gene expression without affecting *Rankl* or *Opg* expression.”

Further specific points:

1–3• Introduction: The authors summarize the role of IL-11 in bone, but do not mention that IL-11 signaling is required for skull formation in mice and also in humans (summarized in PMID: 33940443). This could be added and discussed, as it links IL-11 deficiency to a human syndrome.

We thank the Reviewer for indicating an important point which we did not mention in our original version. According to the reviewer’s suggestion, we mention about the developmental abnormality by missense mutations of IL-11RA gene, causing craniosynostosis in those patients.

Introduction, p. 4, 2 lines from the bottom to p. 5, line 3: “In addition, recent reports demonstrate that IL-11 receptor α knockout (IL-11R $\alpha^{-/-}$) mice, but not IL-11 deficient mice, as well as several coding missense mutations in IL-11RA gene have been linked to craniosynostosis^{11, 12}. Thus, IL-11 receptor signal appears to be associated with abnormalities in human bone development.”

1–4• Fig 1a: There is a scale bar shown, but its length is not defined in the figure legend.

As the Reviewer pointed out, the length of the scale bar was shown only in the original raw picture as 1 mm but not in Figure 1a. According to the reviewer’s comment, we added the size of the scale bar in the legend to Figure 1a as 1 mm.

1-5- Fig 1b,c: The figure legend states that “means \pm SD” is shown here, but only one SD is shown for each genotype.

As pointed out by the Reviewer, we have included \pm SD bars in Figures 1b and 1c.

1-6- Fig 2a: TRAP qPCR levels show high variability, and because of this it is unclear whether it is reduced in IL-11 KO mice. Please repeat the experiment to get a clearer picture.

We appreciate the Reviewer's comment. As recommended by the reviewer, we repeated experiments for TRAP mRNA expression in the femur of WT and IL-11 KO mice. Results demonstrate more clearly that there is no difference in the expression of TRAP mRNA between those mice (Figure 2a).

1-7- Fig 2c: From the text, it is not clear for me why the authors focused on Wnt signaling.

Because there are many signaling pathways that regulate bone formation, our approach may sound arbitrary. However, as explained in the response to the Reviewer's comment 1-2, Wnt pathway is one of the most important signals in the regulation of bone formation, and gene mutations in Wnt signaling pathway in humans cause either high bone mass or severe osteoporosis phenotypes. In addition, we found in our previous study that mechanical loading enhanced IL-11 expression, which inhibited the expression of canonical Wnt signal inhibitors and enhanced Wnt signaling and bone formation in mice (ref. 4). Therefore, the present study is an extension of those previous studies to clarify physiological importance of IL-11 in the regulation of bone remodeling using conventional IL-11 deficient mice. Subsequently, because we found adipose tissue phenotypes in mice with conventional IL-11 deletion, we further created conditional IL-11 deficient mice. We referred to those previous findings (ref. 4) in the Introduction, and explained why we focused on examining the role of Wnt signaling in IL-11-induced bone formation in the present study (Introduction, page 5, line 5 from the bottom to page 6, line 4).

Is this downstream of gp130/Jak/STAT signaling? The connection to IL-11 has to be better presented.

As explained in our response to the Reviewer's comment 1–2, in order to answer these questions, we examined whether IL-11 enhances nuclear translocation of HDAC4/5 and suppresses *Sost* gene expression in the presence or absence of a STAT3 inhibitor. Figures 2h and 2i demonstrate that IL-11 stimulated nuclear translocation of HDAC4/5 which was suppressed in the presence of a STAT3 inhibitor. In Figure 2j, the suppression of nuclear translocation of HDAC4/5 by a STAT inhibitor abrogated the suppression of *Sost* gene expression by IL-11. We added the following sentences to explain those data: Results, p.8, line 9 from the bottom to p.9, line 4: “Because STAT3 is reported to play an important role in regulating osteogenesis²⁰, and because class IIa histone deacetylases, HDAC4 and HDAC5, are shown to be required for loading-induced *Sost* suppression and bone formation²¹, we next examined whether nuclear translocation of HDAC4/5 is enhanced by IL-11 and whether STAT3 inhibition influences nuclear translocation of HDAC4/5 and *Sost* expression in MC3T3-E1 cells. As shown in **Fig. 2h** and **2i**, IL-11 enhanced nuclear translocation of HDAC4/5, and a STAT3 inhibitor, STAT3-IN-1, abrogated IL-11-induced nuclear translocation of HDAC4/5. In parallel with the inhibition of HDAC4/5 translocation, inhibition of STAT3 by STAT3-IN-1 abrogated the suppression of *Sost* expression by IL-11 (**Fig 2j**). These results demonstrate that IL-11 acts on osteoblastic cells via IL-11R α to enhance STAT1/3 phosphorylation, which increases nuclear translocation of HDAC4/5 and suppresses *Sost* gene expression without affecting *Rankl* or *Opg* expression.” Discussion, p.18, lines 4 to 9: “The present results demonstrated that IL-11 enhanced nuclear translocation of class IIa HDAC, HDAC4/5, and a STAT3 inhibitor, STAT3-IN-1, inhibited the nuclear translocation of HDAC4/5 and abrogated the suppression of *Sost* expression by IL-11. These results suggest that IL-11 acts via IL-11R α -gp130-STAT1/3 signaling to enhance translocation of HDAC4/5 and suppresses the expression of *Sost* in the bone.”

1–8· Fig 2e: There is a scale bar shown, but its length is not defined in the figure legend.

Thank you for pointing this out. We indicated in Figure 2e the size of the scale bar as 100 μ m, and mentioned in the legend to Figure 2e.

Furthermore, these pictures have to be quantified (stained cells per field) from an appropriate number of images per mouse and from an appropriate number of animals.

As the Reviewer pointed out, we measured the number of stained cells in each picture. Data are shown in Figure 2e and explained in the legend to Figure 2e.

Legend to Figure 2e (page 40): “Figures in the bottom panel demonstrate numbers of stained cells in each microscopic field at x100 magnification. Data are a mean of a triplicate count in each mouse from 5 mice. *P<0.05, **P<0.01, ***P<0.001 by Student’s unpaired t-test.”

1-9- Fig 4a: The authors show pictures from a single time point, and the effect looks striking; however, it would be more informative to show sections from mice at different ages.

We thank the Reviewer for the suggestion to include time-course changes in HE staining of tibial sections. We performed additional experiments to show the increase in bone marrow adiposity in Figure 4a. We also included a phrase to mention about the timing when the samples were taken, and the size of a scale bar in the legend to Figure 4a.

Legend to Figure 4a (page 41): “(a) H.E. staining of tibia from 4, 8, 12, and 16-week-old WT and IL-11^{-/-} mice. Scale bar = 500 μ m.”

1-10- Line 295-299: “Although reasons for the discrepancy between the systemic IL-11^{-/-} and IL-11R α ^{-/-} mice in bone resorptive activity are not clear, there is a possibility that differences in downstream signaling pathways may exist, as IL-11R α mediates the other signaling pathways than the ones activated by IL-11.” This statement is confusing. IL-11 binds to the IL-11R, which induces gp130 homodimerization and the subsequent activation of intracellular signaling cascades. How should IL-11 and IL-11R induce different signaling pathways? Neither is the IL-11R biologically active without IL-11, nor can IL-11 activate gp130 without IL-11R.

There appeared to be differences in the phenotypes of IL-11^{-/-} and IL-11R α ^{-/-} mice especially in bone resorptive activity, but we were unable to make rational explanation for those discrepancies. Therefore, we tried to speculate if there is any way of explaining the discrepancies. However, as the Reviewer pointed out, our statement in the original manuscript was confusing, and we deleted those statements (Discussion, page 17, the end of the 2nd para).

1-11· Methods: The authors state “IL-11^{-/-} mice appeared normal with similar fertilizing ability (lines 351/352) and “Because these homozygous IL-11-targeted female mice were infertile (lines 390/391).”

Could the authors please comment why these two mouse strains behave differently? IL-11R KO mice have been reported as infertile (PMID: 9679067), so it is interesting to compare the IL-11 KO mice to the IL-11R KO.

We have to apologize that our original statement about fertility of IL-11^{-/-} mice was incorrect. The animal technician who worked with conventional IL-11^{-/-} mice reported no problem about their fertility. After she moved out to a different institute, we stopped breeding IL-11^{-/-} mice, and we wrote in our original manuscript that there was no problem in their fertility without knowing about the problem in fertility. After this issue was pointed out by the Reviewer, we re-started breeding those mice, and found that female IL-11^{-/-} mice were infertile, and we had to cross heterozygous mice to obtain IL-11^{-/-} mice. Thus, both IL-11^{-/-} and IL-11R α ^{-/-} mice were infertile. Therefore, there is a possibility that IL-11 to IL-11R α signaling may be important for fetoplacental development or somewhere during the fetal development. We made changes in our statement in the Methods:

Methods, page 20, lines 3 to 5: “IL-11^{-/-} female mice were infertile, but appeared normal with normal body length and growth curve compared to their wild-type (WT) littermates (Supplementary Fig. 6a,6b).”

1-12· Methods: The authors show the targeting strategy of the two new IL-11 KO strains in the supplement and the results of the PCR are convincing. What is missing is an actual confirmation on the protein level that these mouse do not produce IL-11. This could for example be done by stimulating primary fibroblasts from IL-11 KO and WT littermates with TGF beta and monitoring of IL-11 production.

As a proof to the reduction in IL-11 protein by IL-11 gene deletion, we measured serum IL-11 concentration in control IL-11^{fl/fl}, Apn-Cre;IL-11^{fl/fl} and Ocn-Cre;IL-11^{fl/fl} mice. As shown in Figure 6d, serum IL-11 was markedly reduced in Ocn-Cre;IL-11^{fl/fl}, but not in IL-11^{fl/fl} or Apn-Cre;IL-11^{fl/fl} mice. We believe that these results can be a good in vivo evidence that IL-11 protein production is reduced by bone-specific deletion of IL-11. These results also demonstrate that most of serum IL-11 is derived from bone.

Reviewer #2 (Remarks to the Author):

In this manuscript, the authors report that conventional deletion of IL11 in mice leads to decreased bone mass in two different bones of the skeleton as well as a systemic increase in adiposity and insulin resistance. The decrease in bone density affects both trabecular and cortical bone and is mainly due to decreased bone formation and no significant changes in bone resorption are noted by histomorphometry. The increase in adiposity occurs at multiple site, including the bone marrow, is characterized by an increase in both the number and the size of adipocytes. To determine whether these two phenotypes are independent or linked, the authors then deleted IL11 in the osteoblast lineage using osteocalcin-Cre and in the adipocyte lineage using adiponectin-Cre. Most interestingly, the deletion in osteoblasts recapitulated the entire phenotype whereas the deletion in adipocytes had no measurable effects. PCR analysis of gene expression and genetic analysis of Wnt signaling in vitro and in vivo showed that in all circumstances several Wnt inhibitors were increased in the absence of IL11 in both bone and fat. The authors conclude that IL11 secretion from bone (osteoblasts) acts both as a cytokine, affecting positively bone formation locally, and as a hormone to systemically repress adipogenesis, improve fat metabolism and insulin tolerance. Given that the authors have previously shown that exercise and skeletal loading increase IL11 production in bone, they infer that IL11 from bone is a mechanism by which exercise strengthens bone and increase energy expenditure.

Critique:

Overall, this is a very interesting manuscript reporting novel findings of significant importance in the field of bone, fat and glucose metabolism. The main strength is the identification of IL11 as a potential "hormone" secreted by bone to affect systemically fat and metabolism. In fact, the contrast between the Ocn-cre and the Adipoq-cre phenotypes is very striking and, together with the observed changes in serum levels of IL11, supports well their conclusion. There are however a few weaknesses that, if addressed, would make the manuscript much stronger. In particular:

2-1- The role of Wnt signaling inhibition in IL11-KO mice should be better established, particularly in vivo, possibly by blocking sclerostin and/or Dkk1-2 with antibodies and determining whether the fat and metabolic phenotypes are corrected. Is Wnt inhibited in

adipocytes? Since Wnt are short range molecules, if the changes in fat are Wnt-dependent, it implies that IL11 regulates Wnt signaling locally, in these cells.

We thank the Reviewer for raising a very important question about the role of Wnt signaling in the regulation of adipose tissue mass by IL-11.

In order to answer the question whether the increased adiposity of IL-11 KO mice is counteracted by an inhibition of sclerostin or Dkk1,2 or enhancement of Wnt signaling, we performed additional in vivo experiments using control IL-11^{fl/fl} and Ocn-Cre;IL-11^{fl/fl} mice fed either RD or HFD in the presence or absence of WAY-262611, a β -catenin agonist and a Dkk1 inhibitor. Results are presented in Figure 7a to 7c, and explained in the text.

Results (page 13, 2nd para): “In order to clarify whether the increased adiposity of Ocn-Cre;IL-11^{fl/fl} mice is mediated by a suppression of Wnt signaling in the adipose tissue, we next examined the effect of a β -catenin agonist²² and a Dkk1 inhibitor²³, WAY-262611, on the adiposity of control IL-11^{fl/fl} and Ocn-Cre;IL-11^{fl/fl} mice fed either RD or HFD. Adipose tissue area was larger in mice under HFD than under RD, and was significantly larger in Ocn-Cre;IL-11^{fl/fl} mice than in control mice both under RD and HFD. Treatment with WAY-262611 reduced the adipose tissue area in both control IL-11^{fl/fl} and Ocn-Cre;IL-11^{fl/fl} mice under HFD (**Figure 7a, 7b**). In mice under RD, because the adipose tissue area in vehicle-treated mice was small, WAY-262611 significantly reduced adipose tissue area only in Ocn-Cre;IL-11^{fl/fl} mice (**Figure 7b**). In the adipose tissue, markedly enhanced expression of *CEBP α* in both control and Ocn-Cre;IL-11^{fl/fl} mice under HFD was suppressed by WAY-262611 treatment (**Fig. 7c**). *Dkk2* expression in the adipose tissue was enhanced in Ocn-Cre;IL-11^{fl/fl} mice under both RD and HFD, and WAY-262611 treatment did not affect the expression of *Dkk2*. The expression of a Wnt target gene, *Axin*, was markedly enhanced by WAY-262611 treatment in both control and Ocn-Cre;IL-11^{fl/fl} mice regardless of whether they were under RD or HFD (**Fig. 7c**). *Sost* expression in the bone was enhanced in Ocn-Cre;IL-11^{fl/fl} mice, but was not expressed in the adipose tissue (**Fig. 7d**).

2-2- What evidence do we have that adipocytes express the IL11 receptor and respond to IL11? A nice experiment would be to delete the receptor with Adipoq-cre and determine whether this generates the fat and metabolism phenotype in absence of any bone phenotype. Also, there is no data testing the response of adipocytes to IL11, only deletion is tested.

We again thank the Reviewer for pointing out this important issue.

We agree that we did not have direct evidence that adipocytes express IL-11 receptor and respond to IL-11. Although it is ideal to create *Apn-Cre;IL-11R α ^{fl/fl}* mice and examine whether these mice show only adipose tissue phenotype but normal bone phenotype, such a study requires probably additional two years or so for us, and the time for revision did not allow us to create those mice.

Instead, we decided to create IL-11R α ^{-/-} adipocytes. Because *Dkk1,2* expression is low in most of adipocytic cell lines, we chose C3H10T1/2 cells, because these cells express detectable levels of *Dkk1,2* when adipogenic differentiation is induced. We then created IL-11R α ^{-/-} C3H10T1/2 cells and examined whether IL-11 actions in these cells are abrogated. We presented those data in Figures 7e and 7f, and explained in the text.

Results (Page 13, the last para to Page 14, the last line): “In order to find out whether the increase in the adiposity of *Ocn-Cre;IL-11^{fl/fl}* mice was mediated by a direct effect of IL-11 on adipocytes via IL-11R α , we created an IL-11R α ^{-/-} C3H10T1/2 cell line. When IL-11 was added to WT C3H10T1/2 cells, a dose-dependent suppression of adipogenesis was observed when cells were cultured in an adipogenic medium. In contrast, adipogenic differentiation was enhanced regardless of the presence or absence of IL-11 in IL-11R α ^{-/-} cells (Fig. 7e). In parallel with those observations, mRNA expression of *PPAR γ* and *CEBP α* was suppressed by IL-11 in WT C3H10T1/2 cells, but was markedly enhanced in IL-11R α ^{-/-} cells, and the addition of IL-11 was without effect (Fig. 7f). The addition of IL-11 suppressed the expression of *Dkk1* and 2 in WT cells, whereas the expression of those Wnt inhibitors was markedly enhanced in IL-11R α ^{-/-} cells, and IL-11 was unable to suppress the expression of *Dkk1* and 2 (Fig. 7f). Because *Sost* is not expressed in the adipocytes (Fig. 7d), these results indicate that IL-11 directly acts on the adipocytes via IL-11R α to suppress adipogenesis via enhancement of Wnt signaling by suppressing *Dkk1* and 2.”

Specific points:

2–3. line 142: It is surprising that MC3T3–E1 cells express measurable levels of Soft RNA. This is supposed to define the osteocyte phenotype. How does that compare with osteocyte lines? What number of cycles are required to detect expression?

If MC3T3–E1 cells are cultured in an osteogenic medium for a long time, we are able to observe a time–dependent increase in *Sost* mRNA, at least in the clones of MC3T3–E1 cells we possess. Thus, after a long–term culture, MC3T3–E1 cells differentiate into

osteocytic phenotype, and Sost expression increased to about 40 times as high as that in α -MEM alone. We presented the changes in Sost mRNA in MC3T3-E1 cells cultured in an osteogenic medium in Supplementary Figure 2.

2-4. Fig 2e: the legend does not match

We apologize for our mistake of not including legends to sclerostin and Dkk1,2 staining in Figure 2e.

Figure legend to Figure 2e (Page 40): “(e) X-gal staining and immunostaining for sclerostin, Dkk1 and 2 of tibia from 10-week-old WT and IL-11^{-/-} mice.”

2-5. Fig 3: is fat affected by unloading ? is this change in fat, if any, affected by IL11 deletion?

The reviewer raised an interesting question whether mechanical loading/unloading affected adipose tissue mass. Unfortunately, because the length of unloading/reloading periods was only for 2 to 3 weeks, compared to more than 8 weeks on RD or HFD, we were unable to observe detectable difference in the adiposity of mice under unloading vs loading or WT vs IL-11^{-/-} mice.

Reviewer #3 (Remarks to the Author):

This study identified bone-derived IL-11 as mechano-sensing factor that regulating bone and fat metabolism. Mice with deletion of IL-11 in osteoblasts displayed reduced bone mass with suppressed bone formation and increased systemic adiposity leading to insulin resistance. Furthermore, results showed that the effect of IL-11 on osteoblastogenesis and adipogenesis in IL-11 KO mice mediated via upregulating the Wnt signaling inhibitors. On the other hand, bone formation and adipose tissue mass was not affected in adipocyte-specific IL-11 knockout mice. This study well conducted, but still descriptive and lacking strong supportive data for the mechanism underplaying the function of IL-11 in bone and fat metabolism.

3-1. Does IL-11 acts on adipocyte cells via IL-11Ra ?? More in vitro mechanistic studies needed in adipocyte cell model to identify the direct / indirect effect of IL-11 on adipogenesis. Such studies should investigate the paracrine mode of IL-11 action on adipogenesis.

We agree that we did not have direct evidence that adipocytes express the IL-11 receptor and respond to IL-11. As in the response to Reviewer 2 (2-2), we created IL-11R α ^{-/-} adipocytes. Because Dkk1,2 expression is low in most of adipocytic cell lines, we chose C3H10T1/2 cells, because these cells express detectable levels of Dkk1,2 when adipogenic differentiation is induced. We then created IL-11R α ^{-/-} C3H10T1/2 cells and examined whether IL-11 actions in these cells are abrogated.

We presented those data in Figures 7e and 7f, and explained in the text.

Results (page 13, the last para to Page 14, the last line): “In order to find out whether the increase in the adiposity of Ocn-Cre;IL-11^{fl/fl} mice was mediated by a direct effect of IL-11 on adipocytes via IL-11R α , we created an IL-11R α ^{-/-} C3H10T1/2 cell line. When IL-11 was added to WT C3H10T1/2 cells, a dose-dependent suppression of adipogenesis was observed when cells were cultured in an adipogenic medium. In contrast, adipogenic differentiation was enhanced regardless of the presence or absence of IL-11 in IL-11R α ^{-/-} cells (Fig. 7e). In parallel with those observations, mRNA expression of *PPAR* γ and *CEBP* α was suppressed by IL-11 in WT C3H10T1/2 cells, but was markedly enhanced in IL-11R α ^{-/-} cells, and the addition of IL-11 was without effect (Fig. 7f). The addition of IL-11 suppressed the expression of *Dkk1* and 2 in WT cells, whereas the expression of those Wnt inhibitors was markedly enhanced in IL-11R α ^{-/-} cells, and IL-11 was unable to suppress the expression of *Dkk1* and 2 (Fig. 7f). Because

Sost is not expressed in the adipocytes (Fig. 7d), these results indicate that IL-11 directly acts on the adipocytes via IL-11R α to suppress adipogenesis via enhancement of Wnt signaling by suppressing *Dkk1* and 2.”

3-2. The authors are asked to study the phenotype of newly generated adipocyte specific IL-11Ra KO mice

Although it is ideal to create Apn-Cre;IL-11R α fl/fl mice and examine in vivo whether these mice show only adipose tissue phenotype but normal bone phenotype, such a study requires probably additional two years or so, and the time for revision did not allow us to create those mice. Instead, we created IL-11R α ^{-/-} C3H10T1/2 cells as mentioned in our response to the previous comment 3-1.

3-3. Figure S2, since IL-11 is mainly expressed in bone, what is the serum level of IL-11 in osteocalcin-Cre;IL-11fl/fl mice and adiponectin-Cre; IL-11fl/fl mice ?. These data should be added to clarify that the serum of IL-11 in Adipo-IL-11fl/fl mice still high and therefore the phenotype of adipo-IL-11fl/fl mice is not reflecting IL-11 deletion in adipocyte.

We thank the reviewer for pointing this out. We measured serum IL-11 in Ocn-Cre;IL-11^{fl/fl}, Apn-Cre;IL-11^{fl/fl} and IL-11^{fl/fl} mice. As shown in Figure 6d, serum IL-11 level was not different between Apn-Cre;IL-11^{fl/fl} and IL-11^{fl/fl} mice, but was markedly reduced in Ocn-Cre;IL-11^{fl/fl} mice. We believe that these results confirm our assumption that IL-11 is produced mainly in the bone but not in the adipose tissue. These results also demonstrate that most of serum IL-11 is derived from bone.

3-4. The authors should clarify how their work could provide a novel therapeutic approach to osteoporosis and metabolic syndrome, supplemented with illustrated diagram.

We believe that the present results can explain how the body can adapt physiologically to exercise-induced mechanical loading through mobilization of lipids from adipose tissue as an energy source via the suppression of adipogenesis. We also clarified a new cascade of IL-11 action via the IL-11R α -STAT3 signaling to enhance Wnt signaling via the suppression of *Sost* expression in the bone and *Dkk1*, 2 expression in the adipose tissue.

We added an illustrated diagram to summarize the present observations in Figure 8, and mentioned about the above in the text.

Discussion (page 19, the last para): “In conclusion, the present observations using osteoblasts/osteocytes-specific deletion of IL-11 explains how the body can adapt physiologically to exercise-induced mechanical loading by enhancing the expression of IL-11 in the bone that enhances bone formation in one hand and acts as a hormone in the other hand to reduce adipose tissue mass as an energy source via the suppression of adipogenesis. We also clarified a novel cascade of IL-11 action via IL-11R α -STAT1/3 activation to enhance Wnt signaling via the suppression of Sost expression in the bone and Dkk1, 2 expression in the adipose tissue (Figure 8).”

3–5– Is there any clinical relevance of IL–11 to human bone loss related disease. This study should be supported by clinical data to correlate serum level of IL–11 to bone mass, bone markers, lipid metabolism parameters,etc.

As the reviewer 1 pointed out, developmental abnormality has been reported by missense mutations of IL–11RA gene, causing craniosynostosis in those patients. Thus, at least IL–11RA–mediated signal affects bone development.

In regard to the relationship between serum IL–11 and bone mass, we or others do not have any clinical data to support the present observations. This will be our future project after completing this study. However, as shown in Fig. 3e, serum IL–11 was reduced after tail suspension and returned to the ground control level after reloading in WT mice. These results suggest that serum IL–11 can be used as a diagnostic tool to evaluate whether or not a subject is under sufficient mechanical loading to prevent bone loss and fat accumulation. We added these statements in the text:

Discussion (page 17, lines 3 to 7): “At the same time, because serum IL-11 is reduced by unloading and increased by reloading, increased bone-derived IL-11 in response to exercise acts like a hormone via systemic circulation to control adipose tissue mass. Thus, serum IL-11 may be used to evaluate whether or not a subject takes enough exercise.”

REVIEWERS' COMMENTS

Reviewer #1 (Remarks to the Author):

The authors have addressed all my previous comments and significantly improved the manuscript, which I now recommend for publication.

Reviewer #2 (Remarks to the Author):

Overall, the authors have addressed my comments in a satisfactory manner and I find the manuscript much improved by the revisions and new data provided. Most important, the results now support very well the conclusions and the model proposed, indicating that IL-11 from bone indeed regulates fat tissue and metabolism.

Reviewer #3 (Remarks to the Author):

How the authors made IL-11R α -/- C3H10T1/2 cell line.

This should be explained

RESPONSES TO REVIEWERS' COMMENTS

Reviewer #1 (Remarks to the Author):

The authors have addressed all my previous comments and significantly improved the manuscript, which I now recommend for publication.

[Response]

We thank the Reviewer for giving us important comments to improve our original manuscript.

Reviewer #2 (Remarks to the Author):

Overall, the authors have addressed my comments in a satisfactory manner and I find the manuscript much improved by the revisions and new data provided. Most important, the results now support very well the conclusions and the model proposed, indicating that IL-11 from bone indeed regulates fat tissue and metabolism.

[Response]

We thank the Reviewer for the favorable comment to the revised version of our manuscript.

Reviewer #3 (Remarks to the Author):

How the authors made IL-11R α ^{-/-} C3H10T1/2 cell line.

This should be explained

[Response]

Because we use the same VIKING method to create both IL-11R α ^{-/-} MC3T3-E1 and C3H10T1/2 cells, we combine them in the Methods section under the subtitle “Establishment of *Il11ra* gene knock-out cell lines by VIKING method” in the revised manuscript (page 24 to 25).